# Deep learning detects cardiotoxicity in a high-content screen with induced pluripotent stem cell-derived cardiomyocytes

Francis Grafton[1], Jaclyn Ho[1], Sara Ranjbarvaziri[2], Farshad Farshidfar[1], Anastasiia Budan[1], Stephanie Steltzer[1], Mahnaz Maddah[3], Kevin E Loewke[3], Kristina Green[1], Snahel Patel[1], Tim Hoey[1], Mohammad Ali Mandegar[1]*

[1]Tenaya Theraputics, South San Francisco, United States; [2]Cardiovascular Institute and Department of Medicine, Stanford University, Stanford, United States; [3]Dana Solutions, Palo Alto, United States

**Abstract** Drug-induced cardiotoxicity and hepatotoxicity are major causes of drug attrition. To decrease late-stage drug attrition, pharmaceutical and biotechnology industries need to establish biologically relevant models that use phenotypic screening to detect drug-induced toxicity in vitro. In this study, we sought to rapidly detect patterns of cardiotoxicity using high-content image analysis with deep learning and induced pluripotent stem cell-derived cardiomyocytes (iPSC-CMs). We screened a library of 1280 bioactive compounds and identified those with potential cardiotoxic liabilities in iPSC-CMs using a single-parameter score based on deep learning. Compounds demonstrating cardiotoxicity in iPSC-CMs included DNA intercalators, ion channel blockers, epidermal growth factor receptor, cyclin-dependent kinase, and multi-kinase inhibitors. We also screened a diverse library of molecules with unknown targets and identified chemical frameworks that show cardiotoxic signal in iPSC-CMs. By using this screening approach during target discovery and lead optimization, we can de-risk early-stage drug discovery. We show that the broad applicability of combining deep learning with iPSC technology is an effective way to interrogate cellular phenotypes and identify drugs that may protect against diseased phenotypes and deleterious mutations.

*For correspondence:
mandegar@tenayathera.com

## Introduction

Drug development is a lengthy and expensive endeavor, often requiring an estimated 10 years and $0.8–2.6 billion. The cost—and risk—grows exponentially as drugs advance toward the clinic (*DiMasi et al., 2016*). To reduce costs and risk, pharmaceutical companies need effective screening methods to prevent drug attrition at late stages of the development process.

Major causes of therapeutic attrition are drug-induced toxicity, including cardiotoxicity and hepatotoxicity. Cardiotoxicity alone accounts for approximately one-third of drugs withdrawn due to safety concerns (*Guo et al., 2011*; *Mathur et al., 2013*; *Weaver and Valentin, 2019*). To decrease the potential for toxicity, and for late-stage drug attrition, pharmaceutical and biotechnology industries seek in vitro systems that can identify drug-induced toxicity with phenotypic screening at early stages of development (*Moffat et al., 2014*). This screening enables interrogation of a large number of perturbagens (e.g., small molecules, siRNAs, CRISPR gRNAs) in a target-agnostic assay that measures phenotypic changes (*Eder et al., 2014*).

Some of the best in vitro models rely on human primary cells isolated directly from tissues. These models retain the morphological and physiological characteristics of their tissue of origin (*Kaur and*

*Dufour, 2012*). However, the supply of primary cells is finite. Primary cells also have limited potential for proliferation, are technically complicated to culture for long periods, and are difficult to genetically manipulate. As a result, scientists often turn to immortalized cell lines, such as HEK293T cells, HepG2 human liver cancer cells, and HL-1 cardiac muscle cells. Many of these cell lines have been transformed (*Ahuja et al., 2005*; *Prager et al., 2019*), which may cause karyotypic abnormalities, and they do not fully recapitulate their in vivo counterparts.

More recently, researchers have turned to cell types derived from human induced pluripotent stem cells (iPSCs) (*Takahashi et al., 2007*). Cells derived from iPSCs more closely recapitulate human biology than immortalized cells (*Moffat et al., 2017*; *Scannell and Bosley, 2016*). These derived cells have been used for disease modeling (*Judge et al., 2017*; *Lan et al., 2013*; *Pérez-Bermejo et al., 2020*; *Ribeiro et al., 2017*; *Sharma et al., 2017*; *Sun et al., 2012*), proposed for drug discovery (*Carlson et al., 2013*; *Grskovic et al., 2011*; *Robinton and Daley, 2012*), and used to assess arrhythmogenic and structural liabilities of drugs (*Guo et al., 2011*; *Maddah et al., 2020*). Additionally, iPSCs can be derived from patients carrying deleterious mutations and genetically modified using nucleases (*Judge et al., 2017*; *Kime et al., 2016*; *Mandegar et al., 2016*; *Miyaoka et al., 2012*). They can also be expanded to sufficient quantities and cultured for extended periods that facilitate screening at scale (*Burridge et al., 2014*; *Ghazizadeh and Majd, 2020*; *Sharma et al., 2017*). Hence, iPSC-derived cell types enable high-throughput interrogation and screening using arrayed libraries of perturbagens.

As of early 2021, few studies have used iPSC-derived cells for phenotypic screening in a high-throughput manner that is truly scalable and, most importantly, has an assay readout with an appropriate window for screening and high signal-to-noise ratio. The screening window, commonly assessed using the Z-factor, ensures detection of true-positive hits and limits the number of false-negative and false-positive hits. Additionally, most published studies have been limited to single-readout phenotypes, such as cell survival (*Sharma et al., 2017*; *Sun et al., 2017*), expression of a single marker (*Ghazizadeh and Majd, 2020*), or reporter screens (*McLendon et al., 2017*). While these studies revealed great insight into cardiomyocyte biology, they have not described a general and unbiased screening strategy that can be applied to other cell types and used at scale.

Given the limitation of single-readout assays, researchers have turned to high-content screening to assess cellular features using traditional image processing (*Moen et al., 2019*; *Salick et al., 2020*). However, they must first choose which feature to quantify from a vast array of possible cellular phenotypes, such as nuclear fragmentation, protein aggregation, cell density, organelle damage and distribution, and changes to cytoskeleton organization. While quantifying a combination of these features may reflect the true state or importance of the phenotype, this approach can be challenging. Researchers may need to develop a new algorithm for each feature, and then decide on the biological relevance of each parameter. This strategy is time-consuming and subject to bias.

An alternative approach uses machine learning to decipher cellular features at high accuracies. For example, deep learning, a branch of artificial intelligence, uses a set of machine-learning techniques to train neural networks and learn input data (*LeCun et al., 2015*). Previously, we used phenotypic interrogation with deep learning to identify drug-induced structural toxicity in hepatocytes and cardiomyocytes (*Maddah et al., 2020*). Herein, we aimed to expand that work to induced pluripotent stem cell-derived cardiomyocytes (iPSC-CMs) in a high-content screen (HCS) of 1280 bioactive compounds with putative primary targets. We then sought to validate identified hits using orthogonal assays to assess cellular stress, including mitochondrial respiration and brain natriuretic peptide (BNP) levels. Finally, we aimed to use our deep learning approach to screen a library of 1280 diverse compounds with no known targets and identify chemical structures and frameworks that show a cardiotoxicity signal in iPSC-CMs.

## Materials and methods

**Key resources table**

| Reagent type (species) or resource | Designation | Source or reference | Identifiers | Additional information |
| --- | --- | --- | --- | --- |

*Continued on next page*

*Continued*

| Reagent type (species) or resource | Designation | Source or reference | Identifiers | Additional information |
|---|---|---|---|---|
| Cell line (*Homo sapiens*) | WTC | Gladstone Institutes | | Human iPSC line |
| Antibody | Primary Anti-MYBPC3 Mouse (monoclonal) | Santa Cruz | Sc-137237 | 1:200 |
| Antibody | Primary Anti-ACTN2 Rabbit (monoclonal) | Thermo Fisher Scientific | 701914 | 1:200 |
| Antibody | Secondary Donkey anti-Rabbit IgG (H+L) Alexa Fluor 594 | Thermo Fisher Scientific | A-21202 | 1:500 |
| Antibody | Secondary Donkey anti-Mouse IgG (H+L) Alexa Fluor 488 | Thermo Fisher Scientific | A-21207 | 1:500 |
| Sequence-based reagent | *MYH7* | Thermo Fisher Scientific | Hs01110632_m1 | TaqMan qPCR probe, cardiac marker |
| Sequence-based reagent | *MYH6* | Thermo Fisher Scientific | Hs01101425_m1 | TaqMan qPCR probe, cardiac marker |
| Sequence-based reagent | *TNNI3* | Thermo Fisher Scientific | Hs00165957_m1 | TaqMan qPCR probe, cardiac marker |
| Sequence-based reagent | *TNNI1* | Thermo Fisher Scientific | Hs00913333_m1 | TaqMan qPCR probe, cardiac marker |
| Sequence-based reagent | *MYBPC3* | Thermo Fisher Scientific | Hs00165232_m1 | TaqMan qPCR probe, cardiac marker |
| Sequence-based reagent | *TNNT2* | Thermo Fisher Scientific | Hs00943911_m1 | TaqMan qPCR probe, cardiac marker |
| Sequence-based reagent | *GAPDH* | Thermo Fisher Scientific | Hs99999905_m1 | TaqMan qPCR probe, housekeeping marker |
| Chemical compound, drug | CHIR-99021 | Selleckchem | S1263 | Small molecule |
| Chemical compound, drug | Y-27632 | Selleckchem | S1049 | Small molecule |
| Chemical compound, drug | IWP2 | Sigma | I0536 | Small molecule |
| Chemical compound, drug | Blasticidine S hydrochloride | Thermo Fisher Scientific | 15205 | Small molecule |
| Chemical compound, drug | Bortezomib | Selleckchem | S1013 | Small molecule |
| Chemical compound, drug | Doxorubicin | Selleckchem | S1208 | Small molecule |
| Chemical compound, drug | Givinostat | Selleckchem | S2170 | Small molecule |
| Chemical compound, drug | Bafilomycin | Selleckchem | S1413 | Small molecule |
| Chemical compound, drug | Paclitaxol | Selleckchem | S1150 | Small molecule |

*Continued on next page*

*Continued*

| Reagent type (species) or resource | Designation | Source or reference | Identifiers | Additional information |
|---|---|---|---|---|
| Chemical compound, drug | Cisapride | Selleckchem | S4751 | Small molecule |
| Chemical compound, drug | Sorafenib | Selleckchem | S7397 | Small molecule |
| Chemical compound, drug | FDA-approved Drug Library | Selleckchem | L1300 | Compound library |
| Chemical compound, drug | Discovery Diversity Set | Enamine | DDS-10-Y-10 | Compound library |
| Chemical compound, drug | Dimethyl sulfoxide (DMSO) | Sigma | D8418 | Solvent |

## iPSC culture, iPSC-CM differentiation, and blasticidin selection

WTC iPSCs (*Mandegar et al., 2016*) and derivative lines were maintained under feeder-free conditions on growth factor-reduced Matrigel (BD Biosciences) and fed daily with E8 medium (STEMCELL Technologies) (*Ludwig et al., 2006*). Accutase (STEMCELL Technologies) was used to enzymatically dissociate iPSCs into single cells. To promote cell survival during enzymatic passaging, cells were passaged with p160-Rho-associated coiled-coil kinase (ROCK) inhibitor Y-27632 (10 µM; Selleckchem) (*Watanabe et al., 2007*). iPSCs were frozen in 90% fetal bovine serum (HyClone) and 10% dimethyl sulfoxide (Sigma). iPSCs were differentiated into iPSC-CMs using the Wnt modulation-differentiation method (*Lian et al., 2012*) with 7 µM CHIR (Selleckchem) and 5 µM IWP2 (Sigma) in RPMI media supplemented with B27 (Thermo Fisher Scientific). iPSC-CMs were purified using 1 µM blasticidin (Thermo Fisher Scientific). Then, 15 days after adding CHIR, iPSC-CMs were frozen in 90% fetal bovine serum (FBS) with 10% dimethyl sulfoxide (DMSO). iPSC-CMs were thawed in RPMI with B27% and 10% FBS directly onto Matrigel-coated 384-well plates at a density of 20,000 cells/well. The next day, the media was switched to Tenaya's cardiomyocyte (TCM) media for 6 days before screening. TCM media comprises Dulbecco's Modified Eagle Medium (DMEM) without glucose, 10% dialyzed FBS, 10 mM D-(+)-galactose, and 1 mM sodium pyruvate. To compare TCM against previously published maturation protocols, iPSC-CMs were cultured in RPMI without glucose, supplemented with 4 mM L-lactic acid and 100 nM triiodo-l-thyronine (RPMI/LT3 media) (*Lin et al., 2017*).

## Generation of blasticidin-selectable iPSC line

A plasmid containing the Bsd open reading frame and a selectable marker containing puromycin and green fluorescent protein was knocked into the 3′ end of the *MYH6* locus of WTC iPSCs. Cells were selected with puromycin. After the first round of selection, the selectable marker was removed using a plasmid expressing Cre recombinase (*Figure 1—figure supplement 1A*). Clones were isolated and verified using colony polymerase chain reaction (PCR) to ensure the presence of the Bsd marker and absence of the selectable cassette. Finalized clones showing the appropriate iPSC morphology were sent to WiCell for single-nucleotide polymorphism karyotyping (*Figure 1—figure supplement 1B*).

## Compound library screening

A library containing 1280 bioactive compounds consisting of FDA-approved drugs, tool compounds, and pre-clinical drug candidates was sourced from Selleck Chemicals (Houston, TX). Screening and validation studies on the bioactive compounds were performed at three doses of 0.3 µM, 1.0 µM, and 3.0 µM in 0.1% DMSO. The library of diverse compounds was sourced from Enamine (Kyiv,

Ukraine) and screened at 1.0 μM. StarDrop version 6.6.4.23412 (https://www.optibrium.com/star-drop/) was used to analyze data from the screen of diverse compounds.

## Immunocytochemistry

iPSC-CMs were fixed for 15 min in 4% (v/v) paraformaldehyde (Thermo Fisher Scientific) in phosphate buffered saline (PBS) and permeabilized in 0.1% (v/v) Triton X-100 (Thermo Fisher Scientific) for 15 min. Cells were blocked in 5% (w/v) bovine serum albumin (BSA) (Sigma) with 0.1% (v/v) Triton X-100 in PBS for 60 min. Primary antibodies were diluted in 5% (w/v) BSA and 0.1% (v/v) Triton X-100 in PBS and incubated overnight at 4°C. After treatment with primary antibodies, cells were washed three times in PBS for 15 min each. Secondary antibodies were diluted in 5% (w/v) BSA and 0.1% (v/v) Triton X-100 in PBS, and then incubated for 1 hr at room temperature. After treatment with the secondary antibody, cells were washed three times in PBS for 15 min each. Nuclei were stained using Hoechst 33342 (Thermo Fisher Scientific) (1:1000 dilution). Images were taken using a Cytation 5 microscope (BioTek Instruments) at 10× magnification (nine images per 384-well plate). A list of primary and secondary antibodies with the appropriate dilution is listed in *Supplementary file 4*.

## Sarcomere analysis using scanning gradient Fourier transform

Sarcomere organization and alignment were assessed using scanning gradient Fourier transform (SGFT) with a pattern size of 1.8, scanning resolution of 16, and a Fourier threshold of >0.8 (*Salick et al., 2020*). Briefly, SGFT performs gradient analysis on ACTN2 images to determine the myofibril directionality and then one-dimensional fast Fourier transforms to determine sarcomere organization and alignment. The code for this SGFT algorithm is available on GitHub: https://github.com/maxsalick/SGFT (*Grafton, 2021* copy archived at swh:1:rev:1413b18726172659dcb9a09e6f91653063ab3361).

## Contractility measurements

Contractility video measurements were taken with an SI8000 Cell Motion Imaging System (Sony Biotechnology) and analyzed using Pulse software (*Maddah et al., 2015*).

## Construction of deep learning models and neural network architecture

To avoid complications associated with cell segmentation, we used block image segmentation and seeded a confluent monolayer of iPSC-CMs. Deep learning artificial intelligence models were built using the PhenoLearn platform (https://www.phenolearn.com/). We used PyTorch as the framework for the deep learning library, and a ResNet50 architecture, a 50-layer-deep convolutional neural network. Images from the DMSO control group were trained against the toxic groups (defined as either mildly toxic, toxic, or highly toxic). Each input image was divided into 12 square sub-images to have sizes ranging from 224 × 224 pixels to 300 × 300 pixels (*Maddah et al., 2020*). Each sub-image was flipped and rotated to create seven more augmented sub-images, and then fed into the input layer of ResNet50. Pseudo-image generation by rotation and flipping ensures enough diversity is seen by the network so that the algorithm is not biased based on the orientation of the images (*Moen et al., 2019*). We used 80% of the images to construct the neural network and the remaining 20% to validate the deep learning model. A consistent set of parameters were used for all training operations, including an initial learning rate of 0.01 and 20 epochs. For each training, the final neural network was selected from the epoch with the highest validation accuracy. Z-factor was calculated using the following formula:

$$Z - -factor = 1 - \frac{3(\sigma_p - -\sigma_n)}{|\mu_p - -\mu_n|}$$

## proBNP assay

The proBNP/NPPB human sandwich ELISA kit (Invitrogen) was used to determine the level of secreted human proBNP. Cell culture media was collected from wells containing approximately $10^5$ iPSC-CMs exposed to cardiotoxic drugs for 4 days. Media was diluted to obtain a proBNP concentration between 0.137 and 100 ng/mL (1:3-1:5). A standard mixture of recombinant human proBNP was diluted 1:3 from 0.137 to 100 ng/mL. Then 100 μL of diluted sample and standards were added

to the human proBNP solid-phase sandwich ELISA microplate and incubated at room temperature for 2.5 hr with gentle shaking. The plate was then washed four times with 300 µL of 1× wash buffer. Next, 100 µL/well of biotin conjugate was added to a microplate and incubated at room temperature for 1 hr with gentle shaking. The plate was then washed four times with 300 µL 1× wash buffer. Next, 100 µL/well of streptavidin-horseradish peroxidase solution was added to the microplate and incubated at room temperature for 45 min with gentle shaking. The plate was then washed four times with 300 µL 1× wash buffer. Next, 100 µL/well of TMB substrate was added, and the plate was incubated in the dark at room temperature for 30 min. Then, 50 µL of stop solution was added directly to the TMB substrate and gently mixed. Absorbance was measured using a Cytation 5 microscope at 450 nm and 550 nm. For analysis, background signal was removed by subtracting the 550 nm signal from the 450 nm signal. A linear regression was fit to the standard absorbance measurements. ProBNP values from experimental samples were extrapolated from linear regression. Initial sample dilutions were accounted for when determining the final proBNP concentration.

## RNA extraction and TaqMan qPCR analysis

Approximately $10^5$ to $10^6$ iPSC-CMs were lysed with TRI Reagent (Zymo Research) and frozen at −80°C to ensure complete cell lysis. Total RNA was extracted and washed from lysed cells using the Direct-zol-96 RNA Kit (Zymo Research) according to the manufacturer's instructions. Samples were treated with DNase I for 15 min at room temperature. cDNA was reverse transcribed from 1 µg of RNA through random hexamers using the SuperScript-III kit (Invitrogen) according to the manufacturer's instructions. Real-time qPCR reactions were performed using the TaqMan universal PCR master mix (Applied Biosystems) with the TaqMan probes listed in *Supplementary file 5* (Life Technologies). RT-qPCR reactions were performed using the QuantStudio7 Flex Real-Time PCR systems (Life Technologies). Each reaction was performed in triplicate for a given RT-qPCR run, and each condition had four experimental replicates. Relative expression of the gene of interest was normalized to glyceraldehyde 3-phosphate dehydrogenase (GAPDH) as the housekeeping control using the 2–ΔΔCT method (*Schmittgen and Livak, 2008*).

## High-throughput transcriptional analysis using RNA-seq

Raw RNA-seq reads were aligned with Salmon (*Patro et al., 2017*) (version 1.3.0) to the GENCODE (*Frankish et al., 2019*) version 33 reference transcript assembly (hg38 v.13) using best practice parameters to ensure mapping validity and reproducibility (–seqBias –gcBias –posBias –useV-BOpt –rangeFactorizationBins 4 –validateMappings –mimicStrictBT2). Next, *tximport* (*Soneson et al., 2015*) was used to generate an expression matrix normalized to transcripts per million (TPM). To ensure consistency, we limited our analysis to genes with detectable expression in at least 90% of the samples. Protein-coding genes were determined using Ensembl release 100 human annotation (*Cunningham et al., 2019*) (GRCh38, Apr 2020) and extracted by *biomaRt* (*Durinck et al., 2009*) (version 2.45.9); non-protein-coding and mitochondrial genes were omitted from the analysis. After this step, expression values were re-normalized to TPM. After adding 1 as the pseudo-count, the expression matrix was log$_2$-transformed. For the initial assessment, principal component analysis (PCA) models were generated in the *R* environment using the *prcomp* function, and the first two principal components were used for visualization. To quantify the distances between each drug's samples and generate a PCA-based similarity matrix, we used Euclidean distance in the PCA space, as calculated by the *pca2euclid* function from the *tcR* package. We calculated the distance using all replicates and used the averaged expression in replicates for each drug. Visualizations were generated in R using the *ggplot2* and *ComplexHeatmap* packages.

To compare the differential gene expression analysis of these clusters with the DMSO-treated group, we limited our downstream analysis to the genes in our curated list of genes with high relative expression in cardiac tissue. We also applied the following filters: (1) protein-coding genes (as defined by Ensembl) that had a median log$_2$-transform TPM expression of >1 in at least one cluster and (2) showed a minimum of 0.25-fold change compared to the DMSO control. Transcriptomes in each cluster's replicates were compared with DMSO-treated replicates by Welch's *t*-test. A *t*-statistic value for each gene's expression vector was used to rank-order the transcriptome in each cluster.

To evaluate functional perturbations in each cluster, we limited our analysis to two gene sets: those labeled with a 'cardiac' annotation in MSigDB (*Liberzon et al., 2011*) version 7.2 (127 gene

sets), and the top 100 gene sets that significantly overlapped with our curated cardiac-rich expression gene set as defined by false discovery rate (FDR) q-value. The FDR q-value is the FDR analog of the hypergeometric p-value after correcting for multiple hypothesis testing according to the Benjamini and Hochberg method. In each drug cluster, Welch's t-test statistical values were used to identify and compare most perturbed gene sets to the DMSO-treated group. A p-value < 0.05 was considered significant. We deposited our RNA-seq data on the Gene Expression Omnibus (GEO) database: GEO Submission (GSE172181), https://www.ncbi.nlm.nih.gov/geo/query/acc.cgi?acc=GSE172181.

## Seahorse assay

The Agilent Seahorse XFe96 Analyzer was used to measure mitochondrial function in iPSC-CMs. The 96-well Seahorse plates were coated with Matrigel (1/100 dilution) in a phenol-free medium overnight. WTC iPSC-CMs were seeded at 30,000 cells per XFe96 well and recovered in TCM media for 1 week. Cardiotoxic compounds were diluted to 3 µM, 1 µM, 0.3 µM, and 0.1 µM in 1.0% DMSO in TCM media. Drugs were administered in fresh media for 4 days. Following 4 days of exposure to drugs, the cells were washed and incubated for 1 hr before the assay with Seahorse XF DMEM Basal Medium supplemented with 2 mM glutamine, 2 mM pyruvate, and 10 mM glucose. The Seahorse XFe96 cartridge was prepared according to the manufacturer's guidelines. First, basal oxygen consumption rate (OCR) was measured. Next, the Mito Stress Test was performed with inhibitors injected in the following order: oligomycin (2.5 µM), carbonyl cyanide 4-(trifluoromethoxy) phenylhydrazone (FCCP; 1 µM), and rotenone and antimycin A (0.5 µM). OCR values were normalized to the total nuclear count per well as quantified by Hoechst staining. Basal respiration was calculated as (last rate measurement from the first oligomycin injection) – (minimum rate measurement after rotenone/antimycin A). Maximal respiration was calculated as (maximum rate measurement after FCCP injection) – (minimum rate measurement after rotenone/antimycin A). Spare respiratory capacity was calculated as (maximal respiration) – (basal respiration). ATP production was calculated as (last measurement before oligomycin injection) – (minimum rate after oligomycin injection).

## Proteasome activity assay

Proteasome activity was measured in iPSC-CMs using the Proteasome-Glo Chymotrypsin-Like Assay according to the manufacturer's instructions (Promega). Cells were incubated with the inhibitors (ranging from 2 to 5000 nM) for 1 hr before running the assay. $IC_{50}$s were calculated using PRISM 8 software.

# Results

## iPSC-CM production, selection, and recovery after thaw

To efficiently and reproducibly assess cardiotoxicity in a highly enriched population of cardiomyocytes, we generated a blasticidin-selectable iPSC line by targeting the MYH6 locus (*Figure 1—figure supplement 1A, B*). Following standard differentiation of iPSC-CMs (*Figure 1A*), blasticidin selection reproducibly enriched a heterogenous population of iPSC-CMs to greater than 95% pure iPSC-CMs as measured by ACTN2, TNNT2, and MYBPC3 immunostaining (*Figure 1B, C*). Compared to previously published protocols, recovery in culture media leads to improved maturity metrics based on quantitative polymerase chain reaction (qPCR) markers, contractility (beat rate and velocity), and immunostaining (*Figure 1—figure supplement 1C–F*). To optimize recovery time of iPSC-CMs before screening, daily contractility metrics were performed using Pulse (*Maddah et al., 2015*). Then, 6–8 days after thaw, iPSC-CMs fully recovered and had stabilized contractility readings (*Figure 1D*).

## Known cardiotoxic compounds used to establish the extent of functional damage to iPSC-CMs

To establish training sets for deep learning models, we treated iPSC-CMs with bortezomib (proteasome inhibitor), doxorubicin (topoisomerase inhibitor), cisapride (serotonin 5-HT4 receptor agonist), sorafenib (tyrosine kinase inhibitor), givinostat (histone deacetylase inhibitor), bafilomycin (vacuolar-type H+-ATPase), paclitaxel (microtubule stabilizer), and JQ1 (bromodomain and extraterminal

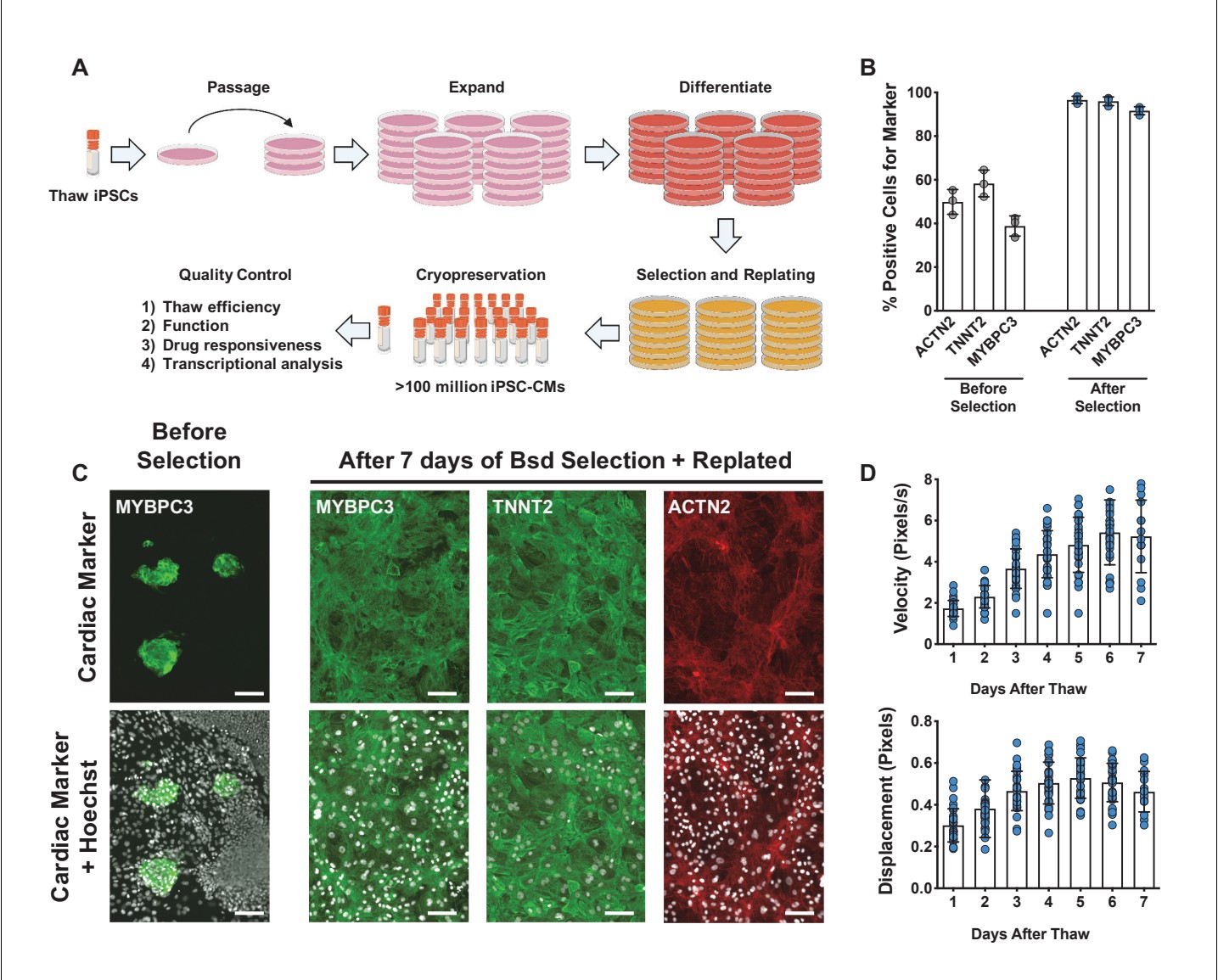

**Figure 1.** Overview of induced pluripotent stem cell-derived cardiomyocytes (iPSC-CM) differentiation, selection, cryopreservation, and recovery for high-content screening. (**A**) Schematic representation of protocol for iPSC-CM differentiation, selection, and cryopreservation for high-content screening. (**B**) iPSC-CMs taken from a representative differentiation stained with ACTN2, TNNT2, and MYBPC3 before and after blasticidin (Bsd) selection. (**C**) Representative immunostaining of an unpurified population of iPSC-CMs before and after selection with blasticidin. iPSC-CMs were seeded as a monolayer for screening. Hoechst stain represented as pseudocolored-to-white for better visualization. Scale bars = 100 μm. (**D**) Blasticidin-purified iPSC-CMs from a representative batch were thawed and recovered for 7 days. Daily contractility metrics were performed to identify the optimal time for recovery of iPSC-CMs after thaw. n = 15–32 technical replicates per day. Error bars = SD.

The online version of this article includes the following source data and figure supplement(s) for figure 1:

**Source data 1.** Overview of iPSC-CM differentiation, selection, cryopreservation, and recovery for high-content screening.

**Figure supplement 1.** Characterization and culturing conditions of WTC-Bsd induced pluripotent stem cell (iPSC) line and differentiated cells.

**Figure supplement 1—source data 1.** Characterization and culturing conditions of WTC-Bsd iPSC line and differentiated cells.

domain inhibitor) (*Lamoureux et al., 2014*). These compounds disrupt a diverse set of cellular pathways, so we included them to ensure that our deep learning models can identify a broad range of cardiotoxic compounds in an HCS.

To determine the level of drug-induced toxicity, we measured beat rate, velocity of contraction, and displacement as a function of dose and duration of exposure (*Figure 2—figure supplement 1*). Results from three representative cardiotoxins (bortezomib, doxorubicin, and bafilomycin) are

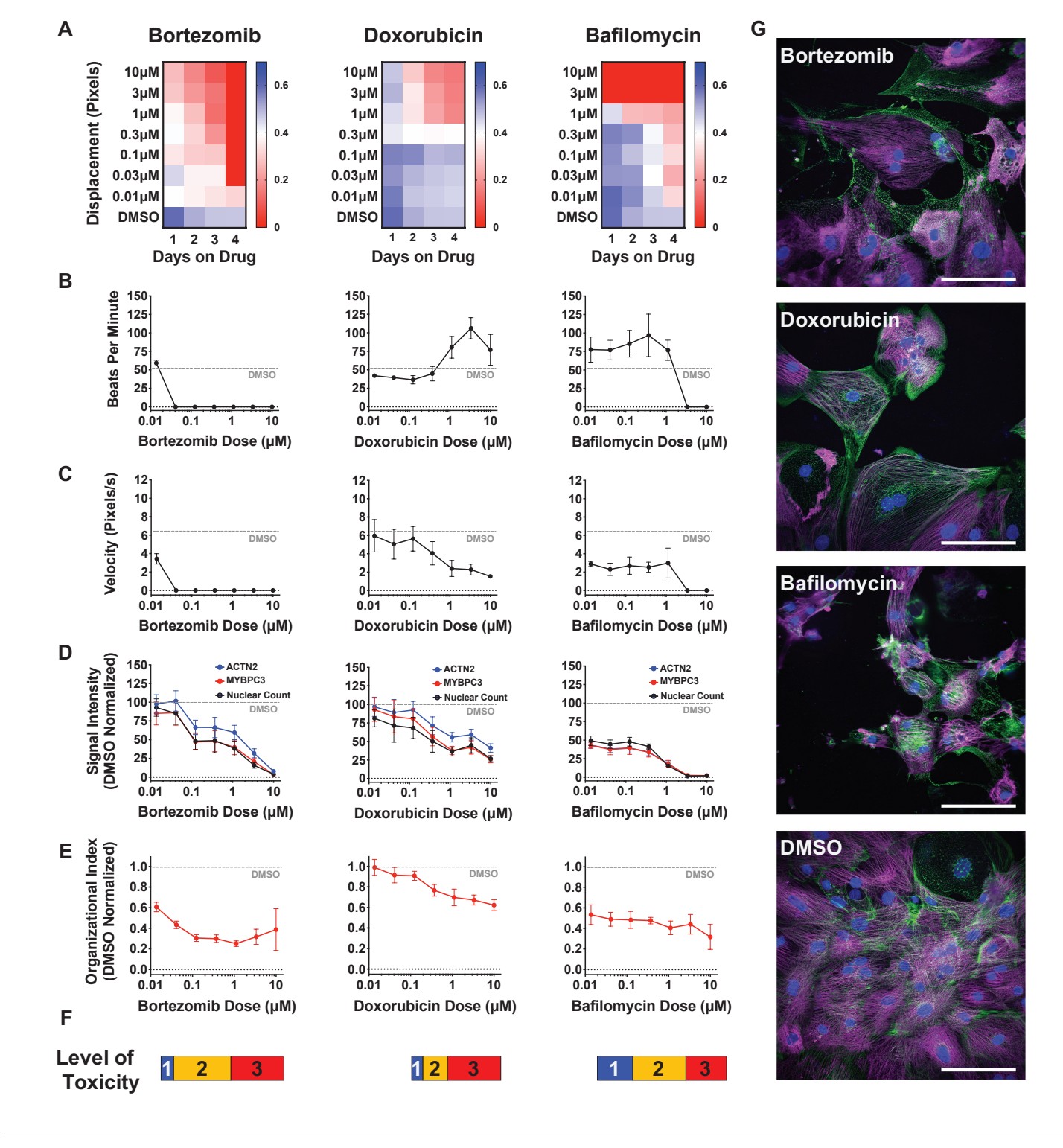

**Figure 2.** Effect of representative cardiotoxins on inducing structural toxicity in induced pluripotent stem cell-derived cardiomyocytes (iPSC-CMs). (**A**) Displacement of iPSC-CMs was measured as a function of dose and duration of exposure to drug. Results from three representative cardiotoxins (bortezomib, doxorubicin, and bafilomycin) are displayed as a heatmap. (**B, C**) Contractility measures of beats per minute and velocity on the fourth day of drug exposure as a function of drug dose. Data indicate a time- and dose-dependent decline in contractility. Error bars = SD; n = 8 technical replicates per drug dose, n = 26 for dimethyl sulfoxide (DMSO) group. (**D**) Signal intensities of sarcomeres stained with antibodies against MYBPC3 and ACTN2 were measured using total fluorescent staining and normalized to DMSO control. Nuclear count was quantified and normalized to DMSO

*Figure 2 continued on next page*

*Figure 2 continued*

control. Dashed gray line, DMSO control condition. Error bars = SD; n = 8 technical replicates per drug dose, n = 26 for DMSO group. (**E**) Relative sarcomere organization quantified using scanning gradient Fourier transform and normalized to DMSO control. The organizational index refers to the average Fourier strength measured across the image. Error bars = SD; n = 8 technical replicates per drug dose, n = 26 for DMSO group. (**F**) Levels of structural toxicity for each drug were binned into the three categories of highly toxic (class 3), toxic (class 2), and mildly toxic (class 1). (**G**) Representative immunostaining of levels of structural toxicity in iPSC-CMs treated with bortezomib (30 nM), doxorubicin (100 nM), and bafilomycin (100 nM). Magenta, MYBPC3; green, ACTN2. Scale bars = 100 µm.

The online version of this article includes the following source data and figure supplement(s) for figure 2:

**Source data 1.** Effect of representative cardiotoxins on inducing structural toxicity in iPSC-CMs.

**Figure supplement 1.** Known cardiotoxins were used to establish levels of functional toxicity in induced pluripotent stem cell-derived cardiomyocytes (iPSC-CMs).

displayed as a heatmap (*Figure 2A*). For each compound, beats per minute and velocity measured on the fourth day of drug exposure are displayed as a function of drug dose (*Figure 2B, C*). We observed that the cardiotoxins had both a dose- and time-dependent effect on contractility measurements. In most cases, the contraction velocity and displacement declined, and the beat rate reduced as a function of dose and time (*Figure 2—figure supplement 1*). In some instances (e.g., doxorubicin), immediately before spontaneous contractility completely stopped, we observed an increased beat rate and reduced contraction velocity and displacement. This change may reflect a compensatory mechanism by which the cells increase beat rate to induce more 'output' while the velocity and displacement are reduced (*Burridge et al., 2016*; *Maddah et al., 2015*).

Next, we assessed sarcomere organization and cell survival. Sarcomere staining intensity was measured using antibodies against MYBPC3 and ACTN2, and cell survival was measured using Hoechst nuclear stain (*Figure 2D*). To assess the structural integrity of the sarcomere and myofibril, which are negatively impacted by cardiotoxin-induced damage (*Burridge et al., 2016*; *Judge et al., 2017*; *Maillet et al., 2016*), sarcomere organization and alignment were also analyzed with the SGFT. This method can quantify subcellular myofibril alignment through one-dimensional fast Fourier transforms following sarcomere mapping (*Salick et al., 2020*; *Figure 2E*). Based on contractility metrics, sarcomeric staining, and nuclear count, we define three levels of structural toxicity binned into the three categories: highly toxic (class 3), toxic (class 2), and mildly toxic (class 1). DMSO-treated cells were designated as non-toxic (class 0) (*Figure 2F* and *Supplementary file 1*). Representative immunostaining of iPSC-CMs stained with MYBPC3 and ACTN2 show various levels of structural toxicity (*Figure 2G*).

## Optimizing the deep learning model

Frozen iPSC-CMs were thawed and allowed to recover in 384-well plates to enable high-content screening. These plates were divided into three categories: a training plate, a validation plate, and library plates. The cells were then immunostained using MYBPC3 before capturing images. Images from the training plate (dosed with known cardiotoxins ranging from 10 nM to 10 µM for 4 days) were used to establish the deep learning models. Images from the validation plate were used to test the accuracy of the deep learning models. And images from the library plates were used to test the toxicity level of compounds. iPSC-CMs in the library plates were exposed to 1280 bioactive compounds at three doses (0.3 µM, 1.0 µM, and 3.0 µM for 4 days) (*Figure 3A*).

Based on defined criteria for reduced contractility, loss of immunostaining, and nuclear count when cells were treated with various doses of cardiotoxins (*Figure 2F* and *Supplementary file 1*), we developed three deep learning models: 4-class, 3-class, and 2-class. The 4-class model distinguished highly toxic, toxic, and mildly toxic compound doses from the non-toxic DMSO-treated condition. The 3-class model combined highly toxic and toxic compound doses and kept mildly toxic compound doses separate. The 2-class model binned highly toxic and toxic compound doses from the non-toxic DMSO-treated condition. The total number of images used per class is outlined in *Supplementary file 2*.

We compared the deep learning accuracies across the three models. All models showed more than 95% accuracy in identifying the non-toxic DMSO-treated condition (class 0). For the 2-class model, the validation showed 100% accuracy in distinguishing the toxic from non-toxic conditions. For the 3- and 4-class models, the accuracies were lower when attempting to distinguish mildly toxic,

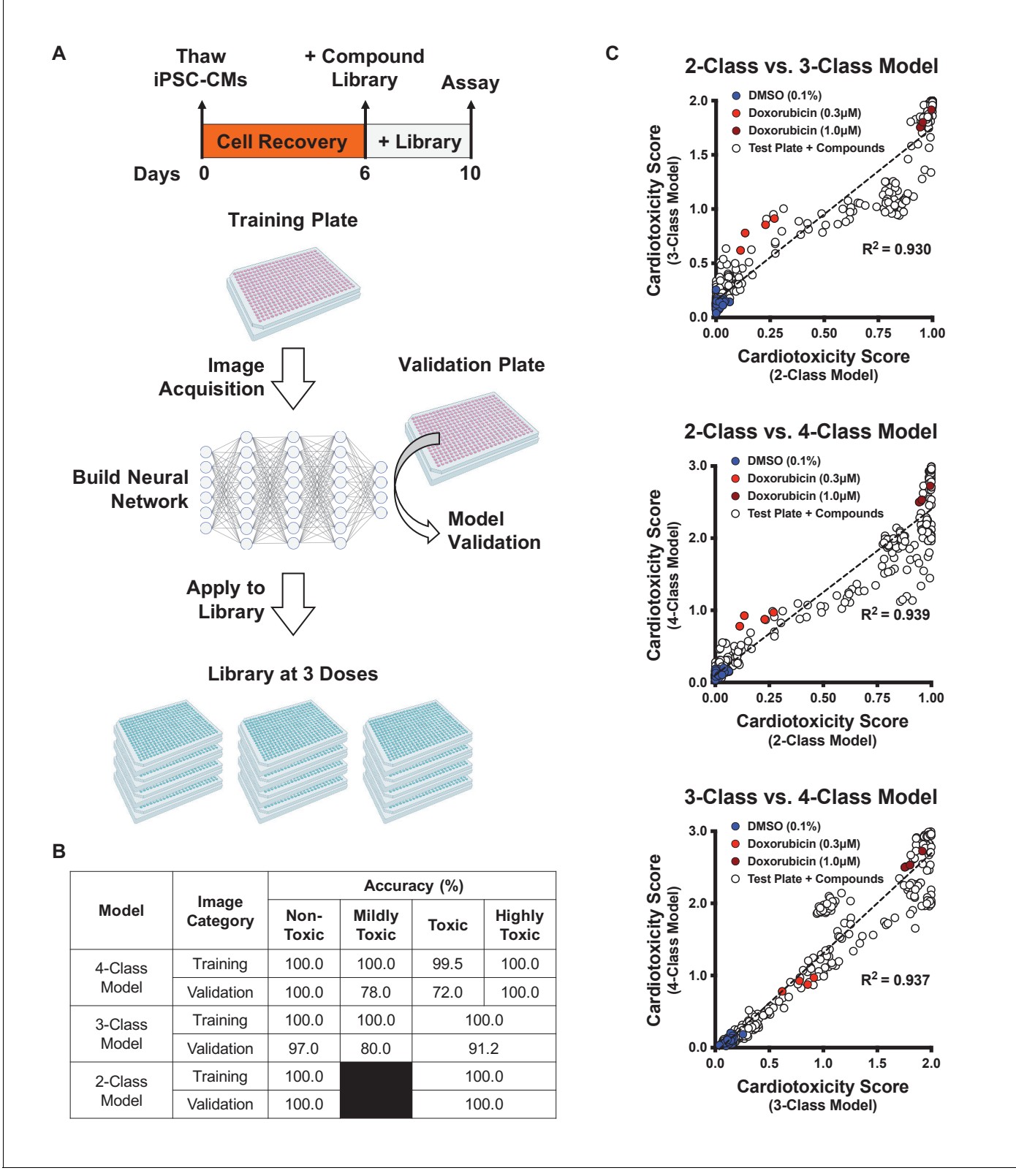

**Figure 3.** Schematic of drug screening and deep learning approach. (**A**) Induced pluripotent stem cell-derived cardiomyocytes (iPSC-CMs) were thawed and allowed to recover for 6 days. Two plates were designated for training and validating the deep learning models. iPSC-CMs were then treated with known cardiotoxins (doses ranging from 10 nM to 10 µM) in the training and validation plates for 4 days. In parallel, a library of 1280 bioactive compounds was added to iPSC-CMs at three doses (0.3 µM, 1.0 µM, and 3.0 µM) for 4 days. (**B**) Deep learning accuracies compared across

*Figure 3 continued on next page*

*Figure 3 continued*

the three models. All models show more than 95% accuracy in identifying the non-toxic dimethyl sulfoxide (DMSO)-treated condition (class 0). Model accuracies were lower when attempting to distinguish mildly toxic, toxic, and highly toxic classes. (C) The three deep learning models compared in a two-dimensional plot on the validation plate. Regardless of how each model was trained (based on the defined images fed into the neural network), all models were strongly correlated when analyzing the validation dataset ($R^2 > 0.93$). The DMSO-treated cells scored the lowest in toxicity, and doxorubicin-treated (1 μM) cells scored the highest. The mildly toxic class (doxorubicin at 0.3 μM) showed an intermediate toxicity separated from the non-toxic and highly toxic classes.

The online version of this article includes the following source data and figure supplement(s) for figure 3:

**Source data 1.** Schematic of drug screening and deep learning approach.
**Figure supplement 1.** Optimization of cell seeding density and model performance.
**Figure supplement 1—source data 1.** Optimization of cell seeding density and model performance.

toxic, and highly toxic classes (*Figure 3B*). This reduced accuracy may be a result of cellular toxicity signatures that follow a continuous spectrum, rather than clearly defined categories.

After training, the three deep learning models were independently used to score the validation plate (which the neural network had not seen before). The results from the three models on the validation plate are compared in a 2D plot in *Figure 3C*. The data suggested that all three models showed a strong correlation upon validation ($R^2 > 0.93$, p<0.0001), regardless of how each deep learning model was trained (*Figure 3C*). As expected, the DMSO-treated cells scored the lowest on the cardiotoxicity scale. Cells treated with 1 μM doxorubicin showed the highest toxicity score, and cells treated with 0.3 μM doxorubicin (the mildly toxic class) showed an intermediate toxicity score separated from the non-toxic and highly toxic class.

## Higher cell seeding densities lead to a higher model accuracy and screening window

We hypothesized that there may be an optimal cell density for screening that leads to best model performance and Z-factor. The Z-factor is a statistical measure of assay variability and reproducibility (*Zhang et al., 1999*). We posited that using a cell density that was too high would tightly pack cells and mask features, whereas using a cell density that was too low would not generate sufficient features for training. Interestingly, as we increased the cell density, we did not observe a bell-shaped effect in the performance of the deep learning models. We found that the higher the cell density, the higher the model accuracy and, most importantly, the higher the Z-factor. At approximately 3000 cells per well of a 384-well plate, the model accuracy and Z-factor plateaued and did not decline with increasing cell density (*Figure 3—figure supplement 1A, B*).

We also found that a minimum cell density is needed to ensure an appropriate screening window. For example, a minimum number of 1500 iPSC-CMs per well of a 384-well plate was required to reach a Z-factor of approximately 0.5, which is an acceptable level for an HCS in cell-based assays (*Figure 3—figure supplement 1B*). Training and validation accuracies were tracked as a function of epochs, or the number of times the entire training dataset passed through the neural network. Two representative cell seeding densities were analyzed, which showed that a higher cell density supports improved model performance (*Figure 3—figure supplement 1C*). Higher cell densities yielded higher cellular content for feature training, and they reduced cell edges and background. In addition, more cell-cell contact between iPSC-CMs led to sarcomere alignment that may reduce heterogeneity in training images. Given these factors, higher cell densities achieved a Z-factor greater than 0.5, which is ideal for HCS (*Figure 3—figure supplement 1D*).

## Deep learning enables detection of toxic compounds in an HCS format

We calculated the dynamic range and Z-factors on the screening plates treated with DMSO (non-toxic control) as well as bortezomib and doxorubicin (cardiotoxic controls) (*Figure 4A*). With enough technical replicates, we saw significant differences in the nuclear count and sarcomere staining intensities (*Figure 4A*). However, when performing an HCS, only one or two technical replicates per test article are used. Thus, using parameters such as nuclear count and staining intensity will not be reliable for identifying hits in high-throughput screening (HTS) (*Figure 4—figure supplement 1A, B*). In this case, the associated Z-factors are far lower than a minimum de facto cutoff of 0.5 for HTS

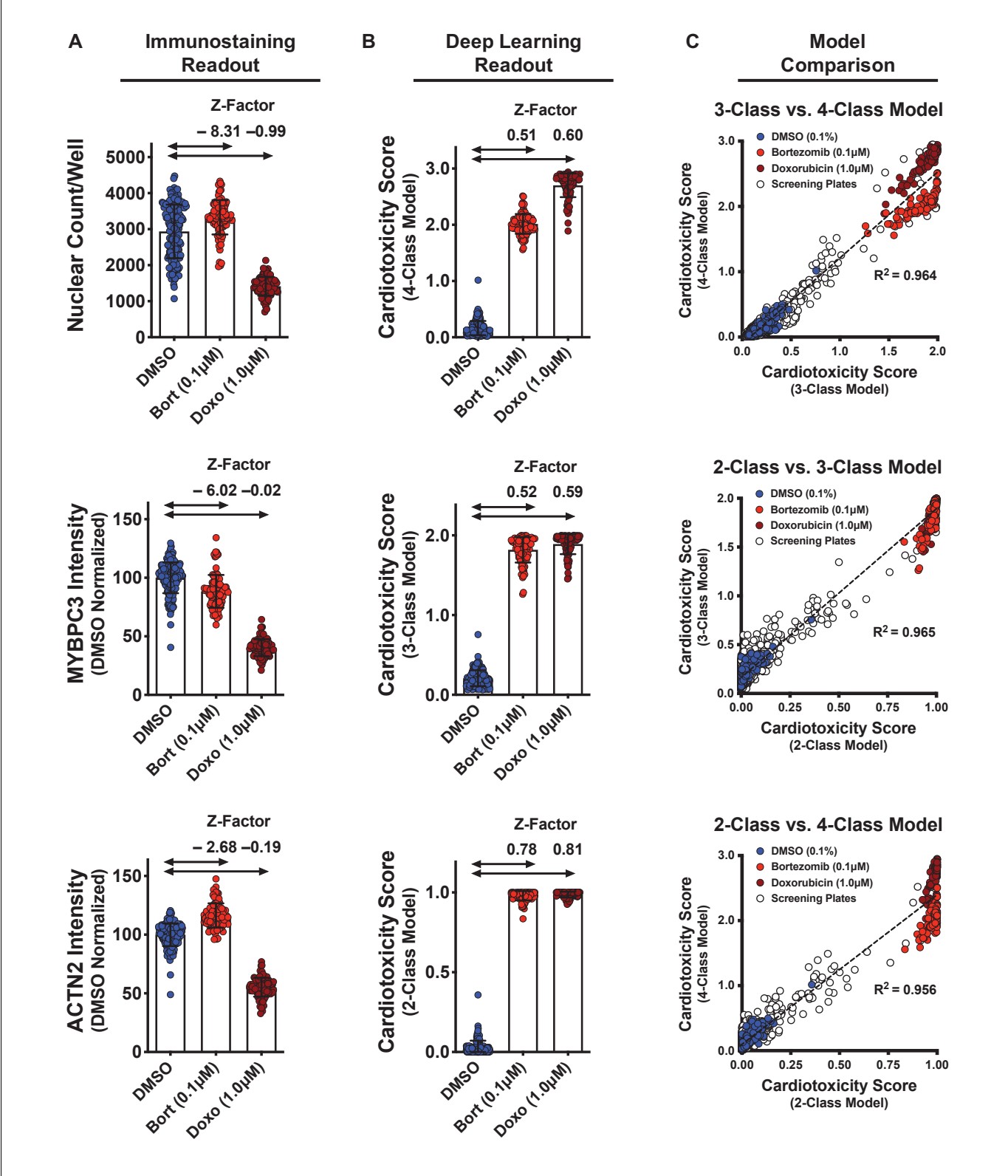

**Figure 4.** Dynamic range and Z-factors of positive and negative controls in screening plates. (**A**) Comparison of Z-factor of nuclear count and MYBPC3 and ACTN2 staining intensities for dimethyl sulfoxide (DMSO) vs. bortezomib (Bort; 0.1 μM) and doxorubicin (Doxo; 1 μM). The Z-factors (< 0) and the limited dynamic range of the nuclear count and sarcomere intensities prevent reliable separation of positive and negative controls. Error bars = SD. (**B**) While using deep learning (regardless of how the models are trained), the dynamic range and Z-factors (> 0.5) enable identification of the toxic controls

*Figure 4 continued on next page*

*Figure 4 continued*

from the DMSO condition. Error bars = SD. (C) Cardiotoxicity scores from all screening wells at three doses of DMSO, bortezomib, and doxorubicin controls are compared in various deep learning models. Regardless of how each deep learning model was trained, cardiotoxicity scores from all three models had strong correlation when applied to the screening plates ($R^2 > 0.95$).

The online version of this article includes the following source data and figure supplement(s) for figure 4:

**Source data 1.** Dynamic range and Z-factors of positive and negative controls in screening plates.

**Figure supplement 1.** Sarcomere intensity and nuclear count on screening plates at three doses.

**Figure supplement 1—source data 1.** Sarcomere intensity and nuclear count on screening plates at three doses.

---

(*Bray and Carpenter, 2017*; *Zhang et al., 1999*). By using deep learning and the appropriate cell seeding density, we can increase the dynamic range and reduce the variability between treatment groups. These changes would increase the screening window coefficient (Z-factor > 0.5) for an HCS (*Figure 4B*).

We compared the cardiotoxicity scores from all screening wells at three doses, including the DMSO, bortezomib, and doxorubicin controls. Regardless of how each deep learning model was trained, all three models showed strong correlation ($R^2 > 0.95$, $p<0.0001$) when applied to the screening plates (*Figure 4C*). All three models also accurately identified the non-toxic DMSO condition and enabled detection of the toxic controls (0.1 µM bortezomib and 1 µM doxorubicin). This result points to the robustness of hit detection and power of deep learning for analyzing large datasets. Given that the 4-class model was trained to separate the toxic and highly toxic classes, this model enables separation between the conditions treated with 0.1 µM bortezomib and 1 µM doxorubicin. To maximize assay sensitivity, it is more valuable to identify mildly toxic compounds than to distinguish highly toxic compounds from toxic compounds (given that highly toxic and toxic compounds show similarly strong phenotypes). To avoid overlooking a mildly toxic compound during the HCS, we proceeded to use the 3-class model to identify and further validate hits.

## Screening using deep learning identifies compounds with cardiotoxic liabilities

We analyzed cardiotoxicity scores by screening 1280 bioactive compounds at three doses using our three deep learning models (*Figure 5—figure supplement 1A, B*). As mentioned, we focused on the 3-class model for further studies and validation rounds (*Figure 5A*). The cardiotoxicity scores from the top 25 hits are displayed as a heatmap from a scale of 0 to 2, such that the most cardiotoxic drugs have scores nearing 2 (red) and the non-toxic class have scores of nearly 0 (blue). For comparison, MYBPC3 and ACTN2 staining intensity and nuclear count are also displayed on a scale of 0 to 1. Most toxic compounds received a score of 1 (red), and least toxic compounds received a score of 0 (blue) (*Figure 5B*). Details about the identified cardiotoxic compounds are listed in *Supplementary file 3*.

A number of identified hits clustered based on mechanism of action and molecular pathways. The top hits showing a cardiotoxicity signal in iPSC-CMs included epidermal growth factor receptor (EGFR) inhibitors (WZ8040, AG-1478), cyclin-dependent kinase 1 (CDK1) inhibitors (BI-2536, PHA-767491), DNA intercalators and synthesis inhibitors (Adrucil, daunorubicin, streptozotocin), ion channel blockers (chlorpromazine, nitrendipine), and other multi-kinase inhibitors (regorafenib). The characterization of hits into target classes is summarized in *Figure 5C*. We classified the protein targets of our top 25 cardiotoxic hits with a PRISM repurposing dataset. Next, we built a protein-protein interaction network using the Search Tool for the Retrieval of Interacting Genes/Proteins (STRING). In the resulting network, we identified seven clusters of protein families, including DNA interactors, ion channel blockers, multi-kinase inhibitors, and cyclin-dependent kinase (CDK) inhibitors (*Figure 5D*).

The top compounds demonstrating a cardiotoxicity signal in iPSC-CMs included several FDA-approved drugs that clinical and pre-clinical studies have linked to adverse cardiovascular events (*Supplementary file 3*). For example, Adrucil (DNA synthesis inhibitor) had a side effect of cardiac toxicity (*RxList, 2020*). Betamethasome (oral glucocorticoid) was identified as a risk factor for heart failure (*Souverein, 2004*). Regorafenib (sorafenib analog, a multi-kinase inhibitor) was linked to increased risk of cardiovascular events in patients with solid tumors (*Chen and Wang, 2018*).

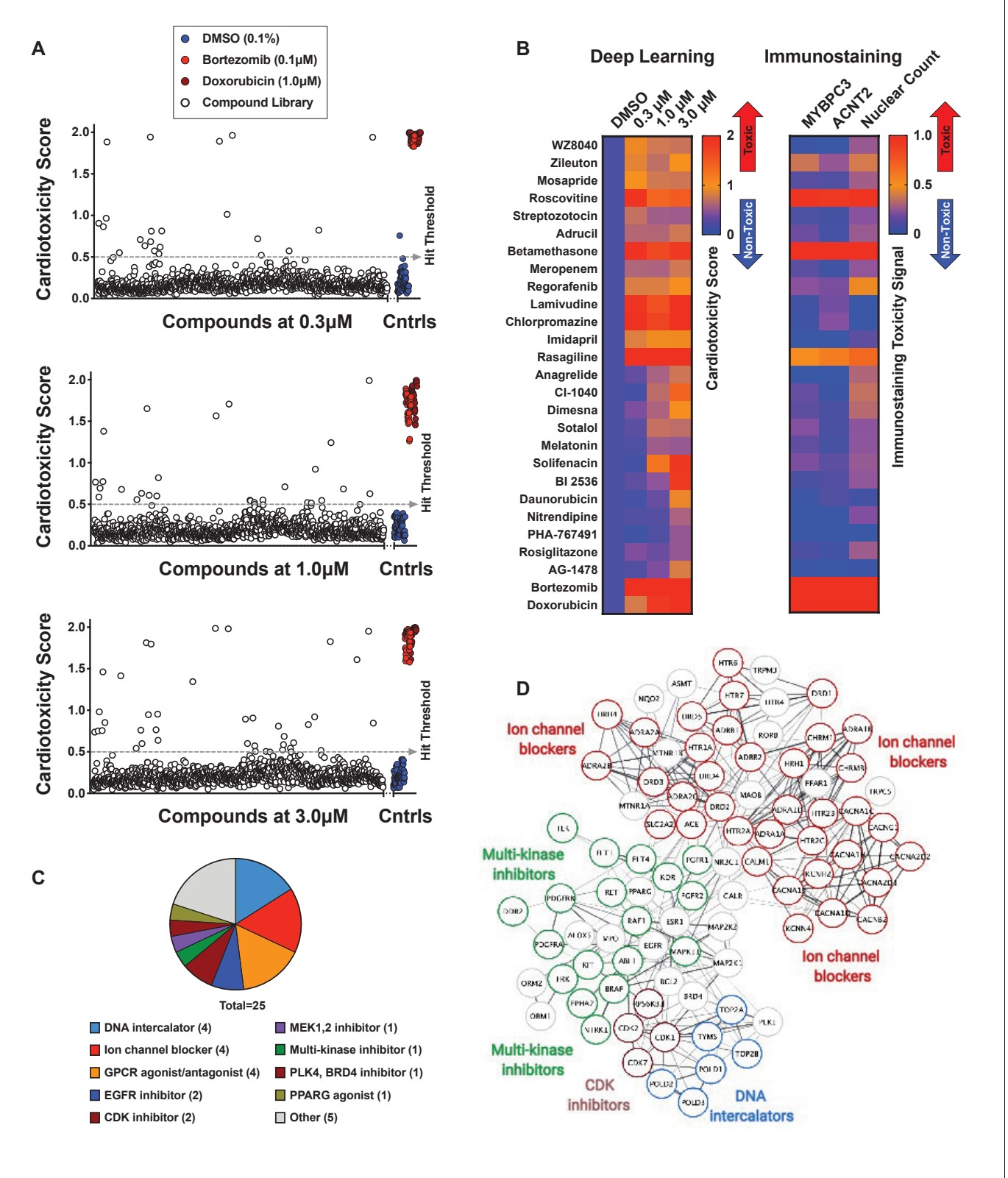

**Figure 5.** Screen of bioactive compound library and identified cardiotoxic hits. (**A**) Cardiotoxicity score from screening 1280 bioactive compounds at three doses. Data from the 3-class deep learning model are plotted. Higher scores correspond to a higher probability of a cardiotoxicity signature. Bortezomib and doxorubicin were used as cardiotoxic controls (Cntrls). (**B**) Cardiotoxicity heatmap scores of top 25 cardiotoxic hits based on deep learning signal. The immunostaining signal is also displayed as a heatmap. Drugs and doses with cardiotoxicity signals in induced pluripotent stem cell-

*Figure 5 continued on next page*

*Figure 5 continued*

derived cardiomyocytes (iPSC-CMs) are indicated in red and yellow, while non-toxic drugs are indicated in blue. Signal intensity from immunostaining and nuclear count was normalized to the dimethyl sulfoxide (DMSO) control and converted to a toxicity scale. Based on the generated heatmaps, phenotypic screening with deep learning is more sensitive in detecting signals than pure immunostaining assays. (C) Target class composition of top cardiotoxic hits from the screen, including DNA intercalators and ion channel blockers, as well as epidermal growth factor receptor (EGFR), cyclin-dependent kinase (CDK), and multi-kinase inhibitors. (D) Search Tool for the Retrieval of Interacting Genes/Proteins (STRING) protein-protein interaction network was used to identify interactions between drug targets (nodes) identified through the PRISM repurposing dataset. Seven clusters in four protein families of DNA (green), multi-kinase (light blue), ion channels (dark blue), and CDK (red) were found based on the highest number of interactions. The minimum required interaction score was set to 0.4, and the edge thickness indicated the degree of data support.

The online version of this article includes the following source data and figure supplement(s) for figure 5:

**Source data 1.** Screen of bioactive compound library and identified cardiotoxic hits.
**Figure supplement 1.** Screening results and hit identification.
**Figure supplement 1—source data 1.** Screening results and hit identification.
**Figure supplement 2.** Deep learning vs. immunostaining analysis.

Chlorpromazine (dopamine and potassium channel inhibitor) was linked to fast and irregular heart rate (*WebMD, 2020*). Anagrelide (PDE3 inhibitor) may cause cardiovascular effects, including fast, irregular, pounding, or racing heartbeat or pulse (*Clinic M, 2020*). Sotalol (beta-blocker and anti-arrhythmic) has serious cardiac side effects, including QT prolongation, heart failure, or broncho-spasm (*FDA.gov, 2020*). Solifenacin (muscarinic receptor antagonist) overdose may cause fast heart-beat (*Plus M, 2020*). Daunorubicin (doxorubicin analog, topoisomerase II inhibitor, and DNA intercalator) causes cardiotoxicity (*Menna et al., 2012*; *Sawyer et al., 2010*). Rosiglitazone (PPAR-γ agonist used to treat patients with type 2 diabetes) was associated with a significant increase in the risk of myocardial infarction (*Nissen and Wolski, 2007*). These reports support the value of screening with deep learning and iPSC-CMs to detect early signs of cardiotoxicity.

## Validation studies using deep learning and orthogonal assays

We further evaluated a subset of hits that showed cardiotoxic liabilities using deep learning analysis but did not show a strong toxic liability using immunostaining image analysis (*Figure 5—figure supplement 2*). This subset included six compounds that showed a cardiotoxic signal in iPSC-CMs: three pre-clinical compounds (WZ8040 [EGFR inhibitor], BI-2536 [CDK inhibitor], and PHA-767491 [CDK inhibitor]) and three FDA-approved compounds (danorubicin [DNA intercalator], nitrendipine [Ca²⁺-channel blocker], and solifenacin [muscarinic receptor antagonist]). As a positive control, we also included tegaserod (5-HT$_4$ agonist), which was withdrawn from the market in 2007 due to adverse cardiovascular effects (*MayoClinic, 2020*). With these seven compounds, we performed a secondary round of validation using deep learning and additional orthogonal assays. This validation showed that all seven compounds had cardiotoxic liability (*Figure 6A* and *Figure 6—figure supplement 1A*).

To further assess those seven validated compounds, we treated iPSC-CMs with the compounds and measured their mitochondrial respiration 4 days later using the Seahorse XFe96 Analyzer. Based on real-time OCR, all seven compounds dose-dependently reduced basal respiration in iPSC-CMs (*Figure 6B* and *Figure 6—figure supplement 1B*), suggesting decreased respiratory function compared to the DMSO control. To indirectly measure OCR, we treated iPSC-CMs with the compounds and oligomycin (ATP synthase [complex V] inhibitor) and then measured their capacity to produce adenosine triphosphate (ATP). Similarly, we found a dose-dependent decrease in the capacity of iPSC-CMs to produce ATP. Finally, we measured whether these compounds affect the spare respiratory capacity (maximum OCR subtracted by basal OCR) in iPSC-CMs. The spare respiratory capacity represents the cell's ability to respond to an energetic stress. When compared to DMSO control, all compounds resulted in a dose-dependent decrease in maximal respiration and spare respiratory capacity (*Figure 6C, D* and *Figure 6—figure supplement 1C*). These data suggest that all seven compounds dose-dependently impair mitochondrial bioenergetics. Thus, by analyzing mitochondrial respiration, we validated the cardiotoxic compounds that were identified by deep learning but not by immunostaining image analysis.

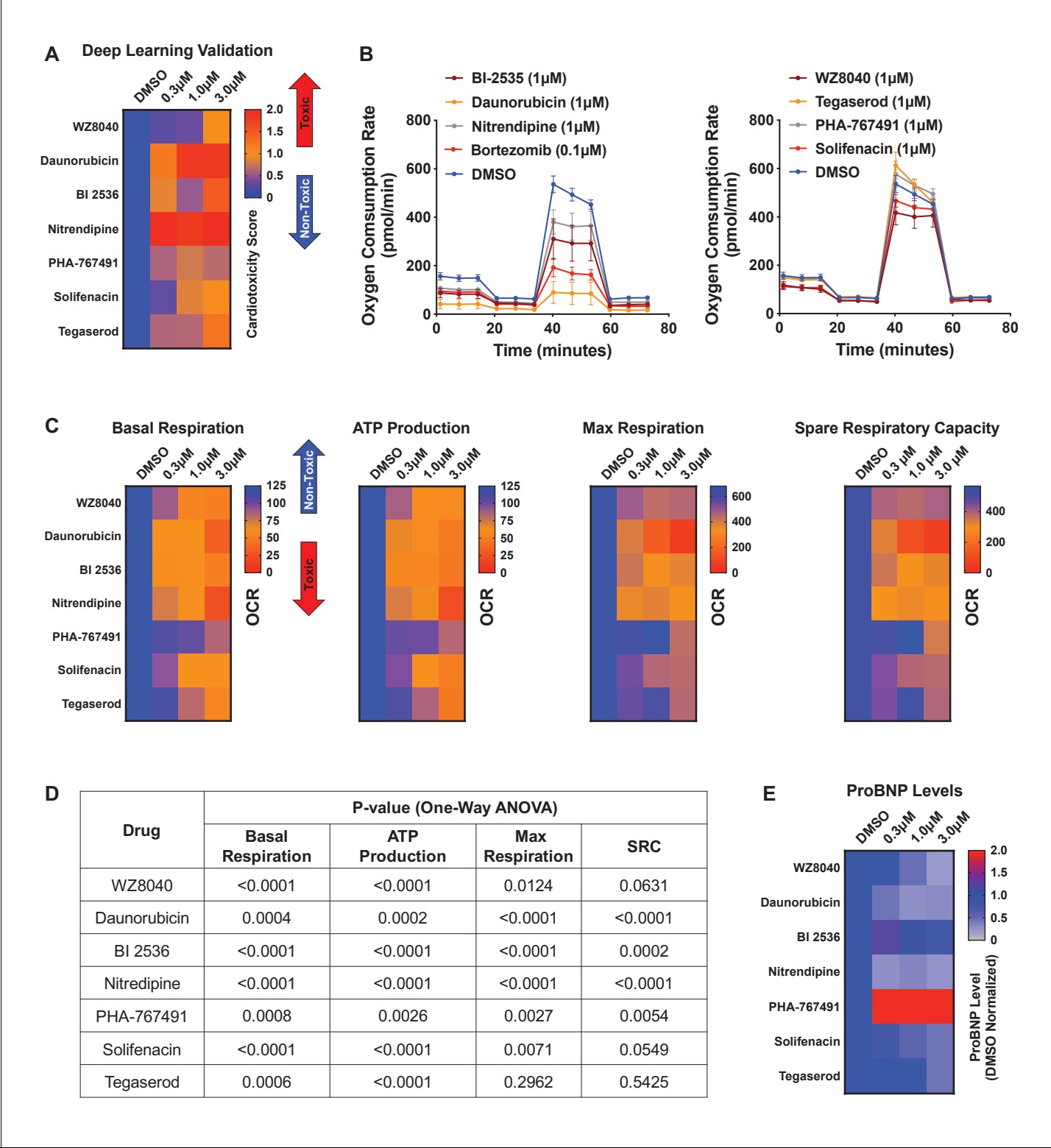

**Figure 6.** Hit validation using deep learning and orthogonal assay analysis. (**A**) Cardiotoxicity heatmap scores for validation of seven compounds: six cardiotoxic hits from the primary screen and tegaserod (known to cause cardiotoxicity and withdrawn from the market). (**B**) Kinetic plots of oxygen consumption rates for the seven compounds and bortezomib as another control. Kinetic data from only the 1 μM dose is plotted. Error bars = SD. (**C**) Heatmaps for basal respiration, adenosine triphosphate (ATP) production, maximal respiration, and spare respiratory capacity. BI-2535, daunorubicin, nitrendipine, and bortezomib had the largest effect on basal respiration, ATP production, and maximal respiration. The oxygen consumption rate (OCR)

*Figure 6 continued on next page*

*Figure 6 continued*

was measured as pmol/min/nuclear count. (D) All seven compounds show significantly different basal respiration and ATP production (one-way ANOVA). All drugs, except tegaserod, show significantly different maximal respiration. SRC: spare respiratory capacity. (E) Heatmaps for ProBNP levels show that PHA-767491 elevates ProBNP levels at all three doses. All other drugs showed no change or a slight decrease in ProBNP levels, suggesting that ProBNP is the least-sensitive marker to assess cardiotoxicity and cellular stress.

The online version of this article includes the following source data and figure supplement(s) for figure 6:

**Source data 1.** Hit validation using deep learning and orthogonal assay analysis.
**Figure supplement 1.** Validation using deep learning and mitochondrial respiration.

## ProBNP levels are not a reliable marker for detecting cardiotoxicity in iPSC-CMs

BNP is natriuretic peptide and hormone secreted from cardiac tissue. During heart failure, BNP secretions are elevated (*Bay et al., 2003*; *Doust et al., 2006*; *Januzzi et al., 2005*). BNP is also expressed in iPSC-CMs, which secrete the hormone into their culture media.

We used a ProBNP ELISA kit as an orthogonal, follow-up assay on media collected from iPSC-CMs treated with the seven validated cardiotoxic compounds and evaluated the levels of BNP. We found an inconsistent pattern of ProBNP levels and cardiotoxicity signal in the media of treated cells. Only PHA-767491 (at all three doses) elevated ProBNP. In contrast, WZ8040, daunorubicin, nitrendipine, and solifenacin dose-dependently reduced ProBNP (*Figure 6E*). This reduction may be due to poor health of iPSC-CMs after treatment with the cardiotoxic compounds. We found that BI-2536 did not alter the levels of ProBNP and that tegaserod only reduced BNP at the highest dose (3 µM) (*Figure 6E*).

## RNA-seq analysis confirms drug-specific changes in key pathways that regulate cardiac muscle contraction, development, and identity

To delineate perturbed transcriptional profiles in iPSC-CMs treated with drugs, we performed RNA sequencing on samples treated with our previously defined nine cardiotoxic compounds (two replicates/drug) and DMSO-treated iPSC-CMs (six replicates). Using PCA, four drugs (bortezomib, daunorubicin, doxorubicin, and BI-2536) fell into three distinct clusters, whereas the other five drugs (nitrendipine, PHA-767491, solifenacin, tegaserod, and WZ8040) clustered more closely with DMSO-treated controls (*Figure 7A*). Based on hierarchical clustering on each sample's projected location in the PCA space with 23 components, we classified the transcriptomic profiles into four clusters (n = 24; cluster 1: bortezomib; cluster 2: doxorubicin and daunorubicin; cluster 3: BI-2536; cluster 4: nitrendipine, PHA-767491, solifenacin, tegaserod, and WZ8040). We generated the heatmap in *Figure 7B* by selecting the top overexpressed and underexpressed genes in each cluster that intersected with a curated set of 310 genes with enriched expression in cardiac tissue. We did not see any major difference in the clustering structure between these two methods. For simplicity, we show the averaged PCA-based similarity matrix for each drug cluster in *Figure 7B*.

Bortezomib-treated cardiomyocytes showed significant downregulation of major genes encoding structural proteins associated with different forms of familial cardiomyopathies (*Haas et al., 2015*). For example, *TCAP* (regulates sarcomere assembly and titin assembly; implicated in familial hypertrophic cardiomyopathy), *MYL3* (myosin light chain 3; implicated in left ventricular hypertrophic cardiomyopathy and restrictive cardiomyopathy), *LDB3*, and *DES* were highly downregulated (*Figure 7C*). Doxorubicin- and daunorubicin-treated clusters were identified by substantial downregulation of *ANKRD1* (associated with dilated cardiomyopathy), *LMNA* (Lamin A/C; known mutations result in cardiomyopathy), as well as *NKX2-5*, and *GATA4* (two transcription factors essential for cardiac development and survival).

BI-2536-related cardiotoxicity in cluster 3 was associated with reduced expression of AGT (angiotensinogen; a critical component of the renin-angiotensin system), PRKCH (protein kinase C, eta; a calcium-dependent serine/threonine protein kinase), and NFATC4 (a cardiac transcription factor required for oxidative phosphorylation activity as well as cardiomyocyte proliferation and differentiation) (*Bushdid et al., 2003*). The fourth cluster with the remaining five drugs (nitrendipine, PHA-767491, solifenacin, tegaserod, and WZ8040) presented more subtle transcriptional changes, including reduced expression of KCNE2 (potassium voltage-gated channel subfamily E regulatory subunit

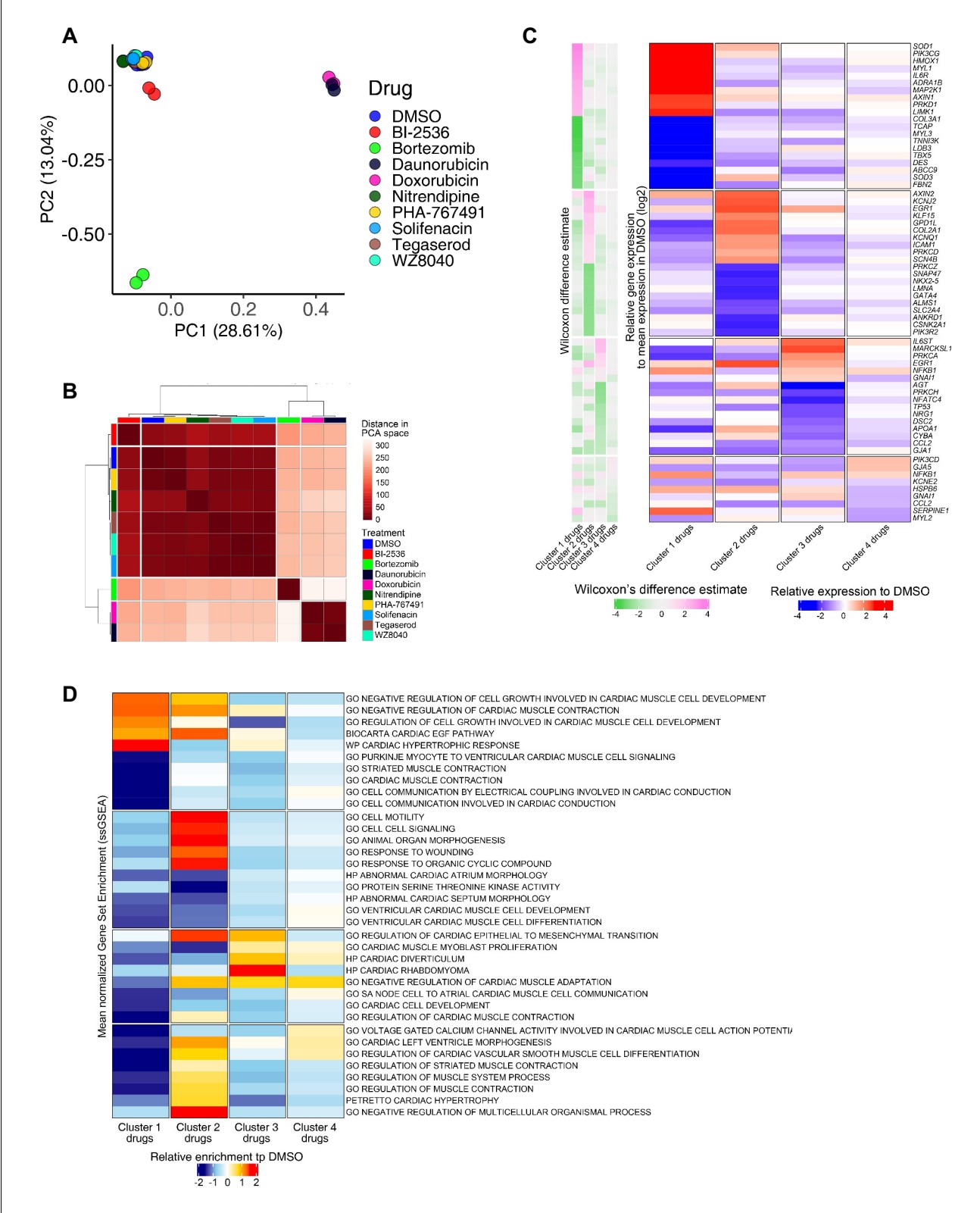

**Figure 7.** RNA-seq analysis of induced pluripotent stem cell-derived cardiomyocytes (iPSC-CMs) treated with candidate cardiotoxic drugs reveals distinct clusters associated with altered gene expression. (**A**) Principal component analysis (PCA) scatter plot representing the extent of transcriptional perturbation compared with dimethyl sulfoxide (DMSO)-treated iPSC-CMs. Overlapping replicates for each drug indicate very high precision of the experiment. Daunorubicin and doxorubicin on principal component 1 (PC1) and bortezomib on PC2 show the most different transcriptional profile

*Figure 7 continued on next page*

*Figure 7 continued*

compared to the DMSO-treated group. Note that daunorubicin and doxorubicin (both DNA intercalators) cluster together. Clustering of other drugs with the control group indicates relatively modest changes when compared with bortezomib, daunorubicin, and doxorubicin. (B) PCA-based hierarchical clustering of similarity matrix shows four distinct clusters of gene expression pattern: bortezomib (cluster 1); daunorubicin and doxorubicin (cluster 2); BI-2536 (cluster 3); and nitrendipine, PHA-767491, solifenacin, tegaserod, and WZ8040 (cluster 4). Cluster 4 shows a gene expression profile closer to the DMSO control group. (C) Heatmap of the most differentially abundant cardiac tissue-enriched genes in each drug cluster versus the DMSO-treated group. Each drug cluster shows up to five most differentially expressed genes (false discovery rate [FDR] < 0.05), showing the presence of four distinct cluster-associated gene groups. (D) Heatmap of the most differentially enriched pathways in each drug cluster. Unbiased single-sample gene set enrichment analysis (ssGSEA) was performed on curated gene lists representing the gene sets most enriched or depleted in each drug cluster as compared with the DMSO-treated group and identified by Welch's *t*-test.

2; associated with arrhythmic abnormalities) and HSPB6 (a molecular chaperone with greater expression in response to stress or tissue damage) (*Li et al., 2017*; *Figure 7C*).

To evaluate functional perturbations in each cluster, we performed single-sample gene set enrichment analysis (ssGSEA) (*Subramanian et al., 2005*) and normalized enrichment score for each sample. Normalized enrichment scores were scaled and centered on the average scores of the DMSO group (*Figure 7D*).

## Deep learning detects structural frameworks that cause cardiotoxicity in iPSC-CMs

To identify chemical structure frameworks that lead to cardiotoxicity in iPSC-CMs, we used our deep learning approach to assess a library of 1280 diverse compounds with no known targets. We used the same conditions as the bioactive library screen. However, to optimize the accuracy and applicability of the diverse library screen, we established a new set of training images and developed a new deep learning model that incorporated all generated datasets. From the 1280 diverse compounds, our deep learning approach identified 33 hits to cause cardiotoxicity in iPSC-CMs (*Figure 8A, B*). Analysis of the structure-activity relationship revealed three structural frameworks with two compounds (also classified as matched pairs) in each framework set (*Figure 8C*). For example, Framework 2a (cardiotoxicity score: 1.41) and its matched pair Framework 2b (cardiotoxicity score: 0.69) showed that adding an ethylene spacer increased the cardiomyocyte toxicity score by twofold (*Figure 8C*). Based on this result, high-content phenotypic analysis using deep learning is a powerful approach to identifying chemical frameworks with cardiotoxic liabilities. By using this approach during the lead optimization process, we can de-risk a clinical development program and potentially reduce drug attrition in late-stage clinical trials.

## Deep learning identifies proteasome inhibitors with reduced cardiotoxicity in iPSC-CMs

Bortezomib and carfilzomib are FDA-approved proteasome inhibitors used for oncology indications, including multiple myeloma and mantle cell lymphoma (*Dimopoulos et al., 2016*; *Richardson et al., 2003*). Patients given these drugs can develop cardiomyopathy, worsening heart rhythm, heart failure, and death due to cardiac arrest (*Moslehi, 2016*). To identify proteasome inhibitors that may reduce cardiotoxic liability, we profiled the potency of 10 proteasome inhibitors using the Proteasome-Glo biochemical assay (*Figure 8D*). For each inhibitor, we calculated the biochemical $IC_{50}$ and measured the cardiotoxicity score using deep learning. We identified six inhibitors with a similar range of biochemical potencies (~12 to ~25 nM): carfilzomib, bortezomib, epoxomicin, delanzomib, ixazomib, and oprozomib. Carfilzomib, bortezomib, and epoxomicin had the highest cardiotoxicity score; delanzomib showed the lowest cardiotoxicity liability; and ixazomib and oprozomib showed medium levels of cardiotoxicity (*Figure 8E*). This analysis is an example of how this type of phenotypic screening would help to de-risk a small-molecule program at an early stage. By focusing on a chemical series (or pathway) that shows high levels of potency for the intended target, we can identify compounds in that series that limit undesired toxicity in a specific cell type with a major liability (such as cardiomyocytes and hepatocytes).

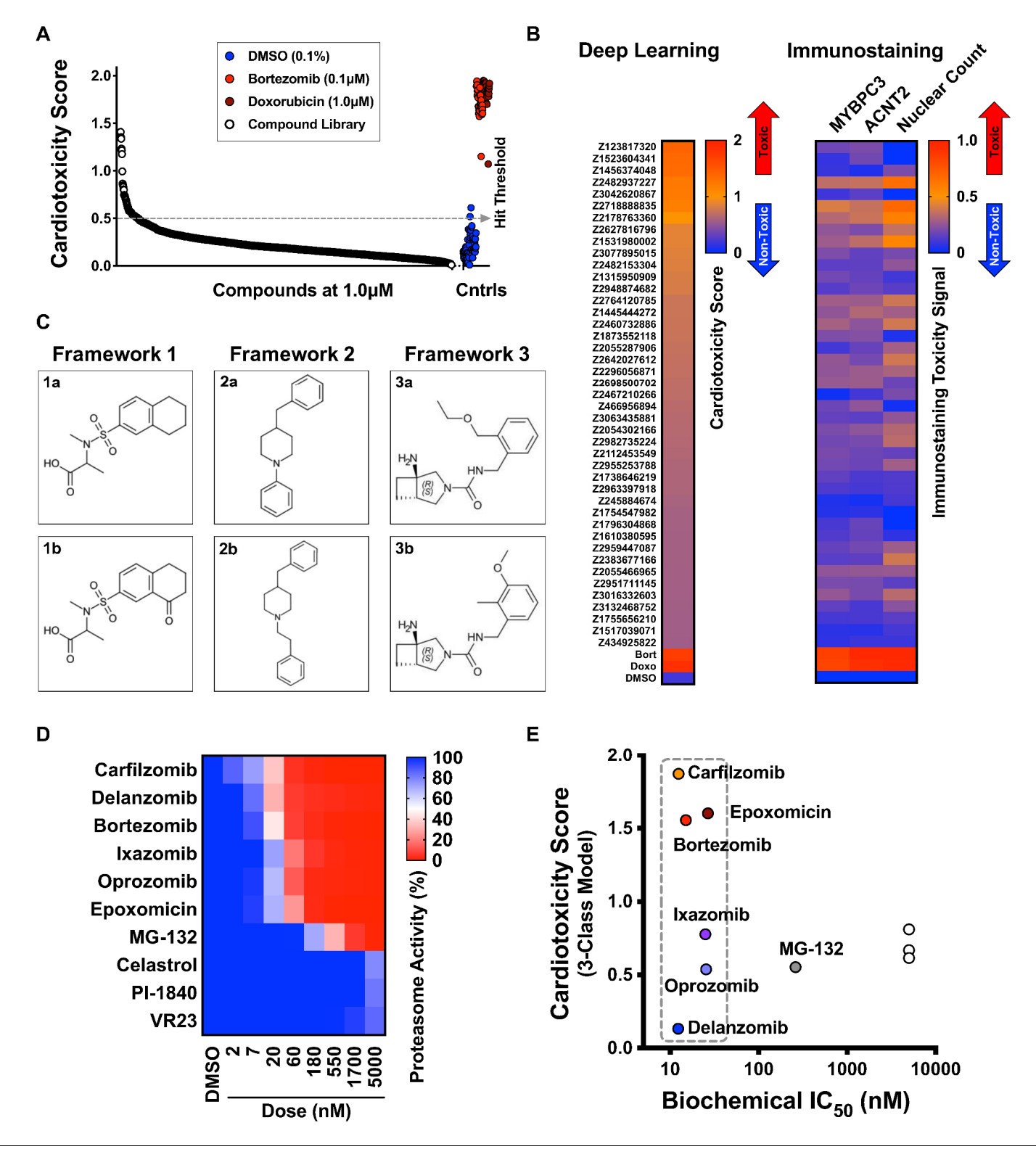

**Figure 8.** Screen of diverse compound library and identified cardiotoxic chemical frameworks. (**A**) Cardiotoxicity scores of 1280 diverse compounds screened at 1 μM. Sorted data from the 3-class deep learning model are plotted. Higher scores correspond to a higher probability of a cardiotoxicity signature. Bortezomib and doxorubicin were used as cardiotoxic controls (Cntrls). (**B**) Cardiotoxicity heatmap scores of top 33 cardiotoxic hits based on deep learning signal. The immunostaining signal is also displayed as a heatmap. Drugs and doses with cardiotoxicity signals in induced pluripotent

*Figure 8 continued on next page*

*Figure 8 continued*

stem cell-derived cardiomyocytes (iPSC-CMs) are indicated in red and yellow, whereas non-toxic drugs are indicated in blue. Signal intensity from immunostaining and nuclear count was normalized to the dimethyl sulfoxide (DMSO) control and converted to a toxicity scale. (C) Three structural frameworks with matched pairs were identified to lead to cardiotoxicity in iPSC-CMs: Framework 1 (1a and 1b, tetrahydronaphthalene-2-sulfonamide core), Framework 2 (2a and 2b, 4-benzylpiperidine core), and Framework 3 (3a and 3b, 3-azabicycloheptane core). (D) Heatmap of chymotrypsin-like biochemical activity of 10 proteasome inhibitors profiled in iPSC-CMs. (E) Cardiotoxicity scores of 10 proteasome inhibitors plotted as a function of chymotrypsin-like biochemical $IC_{50}$. Six of the proteasome inhibitors showed a similar range of biochemical potencies (dashed box). Delanzomib showed the lowest cardiotoxicity score; carfilzomib, bortezomib, and epoxomicin showed the highest cardiotoxicity scores.

The online version of this article includes the following source data for figure 8:

**Source data 1.** Screen of diverse compound library and identified cardiotoxic chemical frameworks.

## Discussion

In this study, we combined the power of human iPSC technology with high-content image analysis using deep learning to detect signatures of cardiotoxicity in a high-throughput chemical screen. First, we treated iPSC-CMs with drugs known to cause cardiotoxicity in the clinic. We combined the contractility readout, immunostaining, and nuclear count to stratify doses of compounds into defined categories: mildly toxic, toxic, and highly toxic. Next, we trained a neural network with fluorescently labeled images to classify the images by degrees of cardiotoxicity with a data-driven method. We then constructed three deep learning models to understand how the categories would differ based on how the toxicities were classified with imaging. We found that all models strongly agreed in identifying hits that cause cardiotoxic liabilities in iPSC-CMs.

Our screening identified classes of compounds that clustered into distinct mechanisms of action and putative target classes. Some of the compounds with the most likely cardiotoxic liabilities included DNA intercalators and ion channel blockers, as well as EGFR, CDK, and multi-kinase inhibitors. We validated these hits using a second round of deep learning and orthogonal assays that measured mitochondrial respiration and evaluated BNP as a marker of stress.

Cardiomyocytes in the adult human heart contain approximately 30% mitochondria by cell volume. These mitochondria critically regulate many cellular processes, such as metabolism, oxidative stress, cell survival, and apoptotic death (*Ventura-Clapier et al., 2011*). The adult human myocardium is highly metabolically active. Approximately 70–80% of its energy derives from oxidative phosphorylation in mitochondria fueled by fatty acid–based oxidative phosphorylation. Conversely, in immature neonatal myocardium or iPSC-CMs, ATP production is predominately through the glycolytic pathway (*Sacchetto et al., 2019*; *Siasos et al., 2018*). In previous studies, exposing iPSC-CMs to various clinically approved compounds with known cardiotoxicities led to a dose-dependent decrease in mitochondrial function in these cells (*Burridge et al., 2016*; *Rana et al., 2012*). Using mitochondrial bioenergetics in iPSC-CMs, we showed that cardiotoxic hits identified from our HCS negatively affected mitochondrial bioenergetics in iPSC-CMs, including reduced basal respiration, ATP production, and maximal respiration rate.

Using RNA-seq analysis, we confirmed that iPSC-CMs treated with drugs identified using our phenotypic cardiotoxicity screening approach resulted in downregulation of key structural and developmental genes, including *TCAP*, *MYL3*, *LDB3*, *DES*, *ANKRD1*, *NKX2-5*, and *GATA4*. In addition, ssGSEA showed downregulation of pathways involved in cardiac muscle contraction, development, and differentiation, further validating our phenotypic screening approach to identifying cardiotoxic drugs.

Use of iPSC-derived models for cardiotoxicity (*Sharma et al., 2017*; *Sirenko et al., 2017*) and hepatotoxicity screening has been proposed (*Mann, 2015*). Phenotypic assays with generic readouts, such as cell viability, are typically less sensitive in identifying subtle toxicity signatures and less likely to enable hit triaging at an early stage. Other toxicity assays in iPSC-CMs, such as electrophysiology and multi-electrode arrays (*Takeda et al., 2018*), are not scalable for HTS. Instead, high-content imaging, such as used in this study, enables researchers to use more complex assays for HTS lead discovery.

Essential components are needed to build a useful deep learning neural network. For example, constructing a relatively large and annotated dataset can strengthen the model. In general, the larger the dataset, the better the performance of the model. In addition, the quality of the dataset

determines the success of the deep learning model. For instance, a more homogenous set of images in well-controlled studies enables better model performance. Conversely, heterogeneity in the cells leads to less accurate and reliable models. Furthermore, the experimenter should ensure that the classes are balanced in the training dataset. Without this balance, the training dataset will display bias toward the most represented class, and in extreme cases, may completely ignore the minority class (*Johnson and Khoshgoftaar, 2019*). In this study, we trained the model by feeding the neural network between 250 and 500 images of pure iPSC-CMs at high cell densities from each class. We also used eight drugs and hundreds of images from each drug to establish our training dataset. This approach ensured that we achieved a relatively large and balanced annotated dataset, while also reducing cellular heterogeneity. In future work in which researchers may aim to identify a cardiotoxicity score with deep learning using in vitro models, we recommend that they establish a deep learning model based on a large training dataset using known cardiotoxins based on clinical data. Then images from test articles can be analyzed through the neural net to compare the level of toxicity to reference cardiotoxic drugs.

Widespread use of deep learning methods in the biological sciences can be challenging. This challenge may be attributed to technical and cultural norms (*Moen et al., 2019*). For example, biological data is inherently complex, heterogeneous, and variable. Thus, a certain threshold of data and computational resources is needed for data annotation to build accurate and useful deep learning models. Another concern has been the 'black box' nature of the method, or not knowing exactly what features the deep learning net is identifying (*Moen et al., 2019*). We propose that the black box is 'a feature and not a bug' of deep learning. We believe the black box allows researchers to interrogate a large set of perturbations in a cell-agnostic, unbiased, and high-throughput manner. Another advantage of deep learning is that deep neural networks significantly outperformed random forest by 22–27% in four measured metrics (*Idakwo et al., 2019*), suggesting that this approach is more robust than traditional chemoinformatics analysis. Although we did not compare deep learning and chemoinformatics in this study, this comparison would be a valuable addition to future research.

In this study, we showcase the power of deep learning in uncovering new biological insights and how the technology can be more accessible to biologists. Deep learning can be used as an additional tool to interrogate subtle and difficult-to-define phenotypes. Our findings support that deep learning is a highly sensitive method to detect cardiotoxicity phenotypes that are undetectable by traditional single-readout assays. While the scope of our study was limited to identifying cardiotoxicity signatures in iPSC-CMs, the method can be applied to identifying toxicity in other cell types.

Screening using iPSC-derived cells with deep learning can also be applied to de-risk early-stage drug discovery and ensure that compounds with on- and off-target toxicity are triaged at an early stage in drug development. In particular, deep learning can be used to identify targets with high-throughput perturbation screening (using small molecules, siRNA, and CRISPR libraries). Our screening strategy is cell and modality agnostic, and it can be applied to a range of indications, such as orphan and neglected diseases. Thus, combining iPSC technology and deep learning can accelerate discovery efforts by supporting rapid model development and interrogation of targets that could be used in different areas of research and for multiple indications.

## Acknowledgements

We thank Crystal R Herron for editing the manuscript. We thank members of Tenaya's drug discovery team for technical assistance and helpful comments on the manuscript.

## Additional information

### Competing interests

Francis Grafton, Jaclyn Ho, Farshad Farshidfar, Anastasiia Budan, Stephanie Steltzer, Kristina Green, Snahel Patel, Tim Hoey, Mohammad Ali Mandegar: is an employee of Tenaya Therapeutics and has stock holdings in the company. Mahnaz Maddah, Kevin E Loewke: is affiliated with Dana Solutions. The author has no other competing interests to declare. The other author declares that no competing interests exist.

## Funding
No external funding was received for this work.

## Author contributions
Francis Grafton, Data curation, Formal analysis, Writing - original draft; Jaclyn Ho, Data curation, Software, Formal analysis, Methodology, Writing - original draft; Sara Ranjbarvaziri, Formal analysis, Writing - original draft; Farshad Farshidfar, Data curation, Software, Formal analysis, Writing - original draft; Anastasiia Budan, Stephanie Steltzer, Data curation, Formal analysis; Mahnaz Maddah, Kevin E Loewke, Software; Kristina Green, Tim Hoey, Resources; Snahel Patel, Data curation, Formal analysis, Validation, Writing - original draft, Writing - review and editing; Mohammad Ali Mandegar, Conceptualization, Resources, Data curation, Software, Formal analysis, Supervision, Funding acquisition, Validation, Investigation, Visualization, Methodology, Writing - original draft, Project administration, Writing - review and editing

## Author ORCIDs
Mohammad Ali Mandegar (iD) https://orcid.org/0000-0002-5323-7891

## Decision letter and Author response
Decision letter https://doi.org/10.7554/eLife.68714.sa1
Author response https://doi.org/10.7554/eLife.68714.sa2

# Additional files

## Supplementary files
• Supplementary file 1. Criteria for Defining Cardiotoxicity Scores and Binning Compounds Used to Establish Deep Learning Models.

• Supplementary file 2. Criteria Used to Construct Classes of Deep Learning Models.

• Supplementary file 3. List of Compounds with Potential Structural Cardiotoxicity and Associated Information.

• Supplementary file 4. Primary and Secondary Antibodies Used for Immunostaining.

• Supplementary file 5. TaqMan qPCR Probes.

• Transparent reporting form

## Data availability
Our RNA-Seq data has been deposited on the Gene Expression Omnibus (GEO) database, GEO Submission GSE172181.

The following dataset was generated:

| Author(s) | Year | Dataset title | Dataset URL | Database and Identifier |
|---|---|---|---|---|
| Grafton F, Ho J, Ranjbarvaziri S, Farshidfar F, Budan A, Steltzer S, Maddah M, Loewke KE, Green K, Patel S, Hoey T, Mandegar MA | 2021 | Deep learning detects cardiotoxicity in a high-content screen with induced pluripotent stem cell-derived cardiomyocytes | https://www.ncbi.nlm.nih.gov/geo/query/acc.cgi?acc=GSE172181 | NCBI Gene Expression Omnibus, GSE172181 |

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
