## [Decision Letter]

**Acceptance summary:**

This study presents a deep learning approach plus iPSC-based high-throughput screening data for risk assessment of cardiotoxicity in the early phase of drug discovery. The authors demonstrate reasonable accuracy using a library of 1,280 bioactive compounds. Overall, this is an interesting study which provides potential tools for risk assessment of cardiotoxicity the early phase of drug discovery if broadly applied.

**Decision letter after peer review:**

Thank you for submitting your article "Deep Learning Predicts Patterns of Cardiotoxicity in a High-Content Screen Using Induced Pluripotent Stem Cell-Derived Cardiomyocytes" for consideration by *eLife*. Your article has been reviewed by 2 peer reviewers, and the evaluation has been overseen by a Reviewing Editor and a Senior Editor. The following individual involved in review of your submission has agreed to reveal their identity: Dr. Feixiong Cheng (Reviewer #2).

The Reviewers and Editors have discussed their reviews with one another, and this decision letter is to help you prepare a revised submission.

Essential Revisions:

1) The article lacks practical discussion, considerations and recommendations on how to validate the cardiotoxicity score when aiming to predict clinical drug cardiotoxicity, which will require comparing with clinical data.

2) This reviewer recommends the authors to read the review manuscript by Walker and colleagues published in 2020 in the journal Archives of Toxicology and entitled "The evolution of strategies to minimise the risk of human drug‑induced liver injury (DILI) in drug discovery and development" to understand how the pharmaceutical industry uses image-based data to predict clinical toxicity. Despite describing applications in liver toxicity, since it focuses extensively on image-based approaches, the practical approaches of the article can be easily translatable to the predictive effort of the neural network method in the cardiotoxicity arena.

3) This reviewer especially recommends considering Cmax or Cmax unbound of tested drugs and investigating IC50 cutoff values for predicting drug safety ranges in relation to the proposed cardiotoxicity score.

4) It is not clear how the deep learning approach outperform traditional approaches, such as doi: 10.1126/scitranslmed.aaf2584.

5) For deep learning models, it is not clear how the authors perform hyperparameter tuning, a key issue for deep learning models.

6) The authors are suggested to compared iPSC-based image-based deep learning models with traditional chemoinformatics approaches, such as random forest-based structure-activity relationships approach.

7) The reviewer is confused why the authors presented ProBNP assays. The authors only show negative results of ProBNP; yet, how it related to the deep learning models in current manuscript.

8) Cardiotoxicity is highly time-dependent and dose-dependent. How the authors address time-dependent and dose-dependent cardio-toxicity in their deep learning models.

9) The authors presented transcriptomics and metabolism analysis. How these omics layers can be integrated into deep learning models in the future studies?

10) The codes and data should be provided in public domains, such as GitHub or other open source websites.

[Editors' note: further revisions were suggested prior to acceptance, as described below.]

Thank you for submitting your article "Deep Learning Predicts Patterns of Cardiotoxicity in a High-Content Screen Using Induced Pluripotent Stem Cell-Derived Cardiomyocytes" for consideration by *eLife*. Your article has been reviewed by 2 peer reviewers, and the evaluation has been overseen by a Reviewing Editor and Matthias Barton as the Senior Editor. The following individual involved in review of your submission has agreed to reveal their identity: Feixiong Cheng (Reviewer #2).

Essential Revisions:

Comment 1– It is difficult to understand how this method (cardiotoxicity score) is predictive of clinical cardiotoxicity if the score is dependent on the training data and drugs used because clinical cardiotoxicity due to drug-induced effects will occur for drugs independently of what would be the surrogate of training data or reference drugs in clinical trials. Probably authors could avoid stating this method as predictive, but instead should present it as an approach for comparing potential cardiotoxic effects between new drugs and reference drugs without predicting clinical safety ranges. A drug can be safe if the range of efficacious concentrations is safely below the toxic range. If the authors still aim to demonstrate that the presented method is predictive of clinical data, a more reflected response to this comment is recommended, taking in consideration the current approaches in the field to test the clinical predictiveness of in vitro data. Since so much clinical and animal data is publicly available for several of the used drugs, it is unconceivable for approaches that aim to be predictive of clinical data to ignore these.

Comments 2 and 3 – There is no proof of principle results or proposed/ described comprehensive strategy or recommendations to predict clinical data from the results derived from this method. In line with the previous comment, the authors seem to be pretending that most of the used drugs are new drugs and that no clinical data is available. The authors also seem to pretend that clinical data is not currently predicted from in vitro cellular data to estimate safety ranges of drug concentration. The authors should rethink their response, especially when considering that some of the used drugs have been used for so long clinically. These are not investigational drugs. If authors want to "pitch" this method as predictive, please add information on how to predict clinical data from it considering how other authors in the field already predict clinical toxic drug effects (safety ranges) from in vitro cellular data. The results from at least one drug should be demonstrated to be predictive and not false-positives or false-negatives. Alternatively, this effort will not be necessary if authors remove from the manuscript any claims that this method is predictive of drug cardiotoxicity.

---

## [Author Response]

Essential Revisions:1) The article lacks practical discussion, considerations and recommendations on how to validate the cardiotoxicity score when aiming to predict clinical drug cardiotoxicity, which will require comparing with clinical data.

In this study, we described the process of deep learning as a methodology. To use deep learning to predict a cardiotoxicity score, we recommend that researchers establish a deep learning model based on a large training dataset using known cardiotoxins based on clinical data (for example, we used eight drugs listed under Supplementary File 1 and Figure 2—figure supplement 1, and hundreds of images from each drug to establish our training dataset). Then images from test articles can be analyzed through the neural net to predict the level of toxicity. We added these details to the Discussion section. If researchers would like to use and/or access our deep learning models and images for their academic or commercial research, we invite them to contact the corresponding author at mandegar@tenayathera.com.

2) This reviewer recommends the authors to read the review manuscript by Walker and colleagues published in 2020 in the journal Archives of Toxicology and entitled "The evolution of strategies to minimise the risk of human drug‑induced liver injury (DILI) in drug discovery and development" to understand how the pharmaceutical industry uses image-based data to predict clinical toxicity. Despite describing applications in liver toxicity, since it focuses extensively on image-based approaches, the practical approaches of the article can be easily translatable to the predictive effort of the neural network method in the cardiotoxicity arena.

We thank the reviewer for this recommendation. We have addressed this suggestion under Comment 3.

3) This reviewer especially recommends considering Cmax or Cmax unbound of tested drugs and investigating IC50 cutoff values for predicting drug safety ranges in relation to the proposed cardiotoxicity score.

Thank you for this recommendation, which would be a valuable part of follow-up studies. We believe that drug toxicity may not be driven only by Cmax but also by AUC exposure (or both may depend on the off-targets). To elucidate which drug’s toxicity is driven by Cmax or AUC, we would need to conduct extensive and regulated follow-up studies based on human data and preclinical species surrogates. However, these studies would be molecule-specific based on analyses such as MetID and not comparable between molecules. Also, data from preclinical species would not be a good indicator for applying Cmax values to human cells. In our paper, the clinical data for the compounds we tested are limited in our libraries (they are composed of limited clinical molecules, preclinical tool molecules, and more). These diverse libraries lack published human pharmacokinetic data. We believe that this analysis would be helpful when more human data is generated and/or released in the public domain.

4) It is not clear how the deep learning approach outperform traditional approaches, such as doi: 10.1126/scitranslmed.aaf2584.

Sharma et al., (2017) evaluated cardiotoxicity mostly by measuring cytotoxicity and contractility (with a follow-up assay using calcium handling and targeted qPCR). To generate data for our deep learning training, we evaluated loss of contractility (Figure 2A-2C and Figure 2—figure supplement 1) and cytotoxicity of eight known cardiotoxins through nuclear count (Figure 2D). We found that measuring cytotoxicity (either through loss of nuclear count or sarcomere intensity measured by MYBPC3 and ACTN2) and contractility did not provide the dynamic range suitable for high-throughput screening. For example, in Figure 4, we show that measuring nuclear count and sarcomere staining intensity did not provide the appropriate Z-factors for hit detection when the methodology is applied in a high-throughput manner.

With deep learning, we achieved Z-factors >0.5, which provided a more reliable method for detecting subtle phenotypic differences with high-throughput screening. For example, a relatively low dose of bortezomib (0.1μM) would not be detected as cardiotoxic when using only cytotoxicity and sarcomere staining. But with deep learning, we can reliably detect patterns of toxicity through sarcomere damage (Figure 4). To clarify this idea, our Discussion section states:

“Our findings support that deep learning is a highly sensitive method to detect cardiotoxicity phenotypes that are undetectable by traditional single-readout assays.”

5) For deep learning models, it is not clear how the authors perform hyperparameter tuning, a key issue for deep learning models.

We built our models using PhenoLearn, without changing the default parameters. For this study, we did not optimize hyper-parameter tuning, but our default values for the parameters were set based on thousands of images and hundreds of trainings we performed on cellular images. We have described some of this methodology in a published study by Maddah et al., 2020, which we referenced in our manuscript.

6) The authors are suggested to compared iPSC-based image-based deep learning models with traditional chemoinformatics approaches, such as random forest-based structure-activity relationships approach.

We thank the reviewers for this comment. In our study, we did not perform traditional chemoinformatics, such as random forest-based structure-activity relationship analysis. We believe this analysis is outside the scope of our manuscript and better suited for a follow-up study. However, we revised our Discussion section to describe the lack of these data as a limitation that would be valuable to assess in a follow-up study. We also referenced a paper by Idakwo et al., 2019 in which deep neural networks significantly outperformed random forest by 22% to 27% in four measured metrics.

7) The reviewer is confused why the authors presented ProBNP assays. The authors only show negative results of ProBNP; yet, how it related to the deep learning models in current manuscript.

ProBNP is a clinical biomarker of cardiac stress. We used a ProBNP ELISA kit as an orthogonal, follow-up assay on media collected from iPSC-CMs treated with the seven validated cardiotoxic compounds and evaluated the levels of BNP. In the media of treated cells, we found inconsistent patterns of ProBNP levels and cardiotoxicity signals. In most cases, the ProBNP signal was lower when treating with a predicted cardiotoxic compound (suggesting loss of cardiac identity). In only one case (PHA-767491) did ProBNP levels increase. Based on our results, we believe that measuring ProBNP levels was not the most reliable marker for predicting cardiotoxicity in iPSC-CMs. We have revised this section in the manuscript for further clarification.

8) Cardiotoxicity is highly time-dependent and dose-dependent. How the authors address time-dependent and dose-dependent cardio-toxicity in their deep learning models.

We thank the reviewers for this insightful point. We agree that cardiotoxicity (or toxicity in general) is both time-dependent and dose-dependent. In our manuscript, we reported that a subset of known cardiotoxic drugs led to a time-dependent and dose-dependent loss of contractility (see Figure 2A and Figure 2—figure supplement 1).

When we applied deep learning to detect cardiotoxicity, we focused on a single timepoint on fixed cells. We did not perform live imaging because this approach, especially on beating cardiomyocytes, is time-consuming and not easily scalable for high-throughput screening. We could perform sarcomere-based live imaging by establishing stable reporter cell lines, or transfecting/transducing with a sarcomeric protein tagged with a fluorescent marker (eg, ACNT2-mKate2 reporter by Judge et al., 2017). However, our study focused on the scalability of the methodology. Thus, for our primary screen, we assessed a single time point and three doses. Also, to identify hits, most primary high-throughput screening is performed at a single dose. In our study, we broadened the dose range by including three doses in the primary screen to address the dose-dependency of cardiotoxicity.

We believe that performing deep learning on live cells over time would be a valuable addition to future work. We used such an approach in previous work (Maddah et al., 2020), in which we used deep learning on live images with a small number of compounds. In this study, we reported both a time-dependent and dose-dependent aspect to detecting patterns of hepatotoxicity (Maddah et al., 2020, Figure 4) and cardiotoxicity (Maddah et al., 2020, Figure 6). We have added some of these details to the Discussion section of our manuscript.

9) The authors presented transcriptomics and metabolism analysis. How these omics layers can be integrated into deep learning models in the future studies?

In our study, we focused on the deep learning score as the primary output to predict patterns of cardiotoxicity, and we used transcriptomics and metabolic studies as orthogonal assays to validate our findings. In this study, we did not have the resources to analyze the transcriptional landscape of enough samples from each drug-treatment group to create a rich transcriptional training dataset as an additional data source for deep learning modeling. When such a dataset is available, the expression matrix can be used along with imaging data as the training dataset, and generated layers can be merged in the subsequent steps. Such a dataset could also be analyzed separately by either deep learning or traditional machine learning methods, and the generated classifier can be provided to create an Ensemble model in the next step. In this approach, multivariate projection methods (e.g., methods based on partial least squares, multivariate regression methods, random forests) and Support Vector Machines could be used to develop transcriptional classifiers.

10) The codes and data should be provided in public domains, such as GitHub or other open source websites.

The code for the SGFT algorithm by Salick et al., 2020 is available on GitHub:

https://github.com/maxsalick/SGFT

(We added this link to our Materials and methods)

Our RNA-Seq data has been deposited on the Gene Expression Omnibus (GEO) database. GEO Submission (GSE172181):

https://www.ncbi.nlm.nih.gov/geo/query/acc.cgi?acc=GSE172181

(We added this link to our Materials and methods)

The PhenoLearn code is proprietary. However, we provided information on how the code is constructed in the Materials and methods section:

“We used PyTorch as the framework for the deep learning library, and a ResNet50 193 architecture, a 50-layer-deep convolutional neural network. Each input image was divided into 12 square sub-images to have sizes ranging from 224 x 224 pixels to 300 x 300 pixels (Maddah et al., 2020). Each sub-image was flipped and rotated to create seven more augmented sub-images, and then fed into the input layer of ResNet50. Pseudo-image generation by rotation and flipping ensures enough diversity is seen by the network so that the algorithm is not biased based on the orientation of the images (Moen et al., 2019). We used 80% of the images to construct the neural network and the remaining 20% to validate the deep learning model. A consistent set of parameters were used for all training operations, including an initial learning rate of 0.01 and 20 epochs.”

[Editors' note: further revisions were suggested prior to acceptance, as described below.]

Essential Revisions:Comment 1– It is difficult to understand how this method (cardiotoxicity score) is predictive of clinical cardiotoxicity if the score is dependent on the training data and drugs used because clinical cardiotoxicity due to drug-induced effects will occur for drugs independently of what would be the surrogate of training data or reference drugs in clinical trials. Probably authors could avoid stating this method as predictive, but instead should present it as an approach for comparing potential cardiotoxic effects between new drugs and reference drugs without predicting clinical safety ranges. A drug can be safe if the range of efficacious concentrations is safely below the toxic range. If the authors still aim to demonstrate that the presented method is predictive of clinical data, a more reflected response to this comment is recommended, taking in consideration the current approaches in the field to test the clinical predictiveness of in vitro data. Since so much clinical and animal data is publicly available for several of the used drugs, it is unconceivable for approaches that aim to be predictive of clinical data to ignore these.Comments 2 and 3 – There is no proof of principle results or proposed/ described comprehensive strategy or recommendations to predict clinical data from the results derived from this method. In line with the previous comment, the authors seem to be pretending that most of the used drugs are new drugs and that no clinical data is available. The authors also seem to pretend that clinical data is not currently predicted from in vitro cellular data to estimate safety ranges of drug concentration. The authors should rethink their response, especially when considering that some of the used drugs have been used for so long clinically. These are not investigational drugs. If authors want to "pitch" this method as predictive, please add information on how to predict clinical data from it considering how other authors in the field already predict clinical toxic drug effects (safety ranges) from in vitro cellular data. The results from at least one drug should be demonstrated to be predictive and not false-positives or false-negatives. Alternatively, this effort will not be necessary if authors remove from the manuscript any claims that this method is predictive of drug cardiotoxicity.

We thank the reviewers for their insightful comments. We removed all claims that this method predicts clinical cardiotoxicity from the manuscript. We rephrased all relevant parts of the manuscript to indicate that this method can be used to detect/identify a cardiotoxicity signal in an in vitro system using induced pluripotent stem cell-derived cardiomyocytes. Also, we took the reviewer’s recommendation and modified part of our discussion to indicate that this method can be used to compare the level of toxicity to reference cardiotoxic drugs.